# Stabilizing a mammalian RNA thermometer confers neuroprotection in subarachnoid hemorrhage

Min Zhang [1,2,3,25] ✉, Bin Zhang [4,5,6,25], Chengli Liu [7,25], Marco Preußner [1], Megha Ayachit [8], Weiming Li[9], Yafei Huang[10,11], Deyi Liu[12], Quanwei He[13], Ann-Kathrin Emmerichs[1], Stefan Meinke [1], Shu Chen[14], Lin Wang[15], Liduan Zheng[16], Qiubai Li [17,18], Qin Huang[19], Tom Haltenhof[1], Ruoxi Gao[20], Xianan Qin [21], Aifang Cheng[22,23], Tianzi Wei[24], Li Yu [3], Mario Schubert [8], Xin Gao [4,5,6], Mingchang Li [7] ✉ & Florian Heyd [1] ✉

Mammals tightly regulate their core body temperature, yet how cells sense and respond to small temperature changes remains incompletely understood. Here, we discover RNA G-quadruplexes (rG4s) as key thermosensors enriched near splice sites of cold-repressed exons. These thermosensing RNA structures, when stabilized, mask splice sites, reducing exon inclusion. Specifically, rG4s near splice sites of a cold-repressed poison exon in the neuroprotective RBM3 are stabilized at low temperatures, leading to exon exclusion. This enables evasion of nonsense-mediated decay, increasing RBM3 expression at cold. Importantly, stabilizing rG4 through increasing intracellular potassium with an FDA-approved potassium channel blocker, mimics the hypothermic effect on alternative splicing, thereby increasing RBM3 expression, leading to RBM3-dependent neuroprotection in a mouse model of subarachnoid hemorrhage. Our findings unveil a mechanism how mammalian RNAs directly sense temperature and potassium perturbations, integrating them into gene expression programs. This opens new avenues for treating diseases arising from splicing defects and disorders benefiting from therapeutic hypothermia, especially hemorrhagic stroke.

In most mammals, including humans, core body temperature is rigorously regulated to stay around 37 °C, with minor circadian fluctuations[1]. Deviations from normothermia lead to severe physiological disruptions, with accidental hypothermia (< 35 °C) impairing systemic functions[2] and extreme hyperthermia (> 40.5 °C) causing heatstroke[3]. While therapeutic hypothermia (TH) is a recognized neuroprotective strategy[4], such as in intracerebral and subarachnoid hemorrhage (SAH)[5-8], the molecular mechanisms underlying TH and strategies for sustained neuronal protection remain unclear[9]. Recently, the cold shock protein RBM3 (RNA-binding motif 3) has been identified as a key mediator of the neuroprotective effects of TH[10,11].

At the molecular level, cellular responses to temperature fluctuations may involve multiple mechanisms in mammals. Even minor shifts (~1 °C) influence RNA splicing by modulating SR protein phosphorylation in many different cell types. This regulation, mediated by CDC-like kinase (CLK), depends on temperature-induced conformational rearrangements within its active site[12]. Beyond protein-mediated regulation, RNA itself may function as a temperature sensor due to its structural flexibility. While RNA thermo-sensing is well established in bacteria[13], yeasts[14], and plants[15-17], its role in response to the small physiological temperature changes remains largely unexplored in mammals.

RNA G-quadruplexes (rG4s) represent comparably stable secondary structures formed by Hoogsteen guanine base pairs (G-G). Notably, RNA G-rich elements, which can be stabilized by G4 ligands, such as PDS[18], Phen-DC3[19], and potassium ions[20], were suggested to form rG4s in vitro and in cells[21]. They play crucial roles in diverse biological processes, contributing to multiple molecular functions, such as alternative polyadenylation, phase separation of RNA granules, and translation[22,23]. Furthermore, genome-wide analysis in mammalian cells found an enrichment of G-rich DNA elements[24] and DNA G4 structure[25] near splice sites. While G4 elements are prevalent in the genomes of eukaryotes and can form robust structures in vitro, there is ongoing debate regarding rG4 folding within mammalian cells[21,26]. Importantly, the involvement of rG4s as physiologically dynamic, temperature-dependent regulatory RNA structures in mammals has not been investigated.

Here, we identify an enrichment of rG4 motifs near splice sites of cassette exons repressed at lower temperatures (~33 °C) in mammals. Exclusion of these exons upon cold shock correlates with rG4 scores, and rG4s form more readily at lower temperatures and elevated intracellular potassium levels. Among the cold-repressed exons with rG4s surrounding splice sites, a poison exon in RBM3 (exon 3a) is a key responder during TH. Using RBM3 exon 3a as a model, we show that increasing intracellular potassium ion through the broad voltage-gated potassium channel blocker 4-aminopyridine (4-AP) stabilizes rG4s, leading to RBM3-dependent neuroprotection in a subarachnoid hemorrhage mouse model. This highlights rG4s as RNA thermometers in mammalian cells that modulate alternative splicing and provides evidence for a novel neuroprotective mechanism of 4-AP.

## Results

### Significant enrichment of rG4 motifs around splice sites of cold-repressed cassette exons

We reasoned that rG4 structures would be well-suited to act as physiological thermosensors in mammalian cells (Fig. 1a). To inspect this possibility, using our previously published RNA-seq data in HEK293T cells at different temperatures[12], we calculated the G-content for sequences surrounding splice sites of cassette exons, using upstream and downstream constitutive exons as controls (Fig. 1b). We classified the cassette exons into three categories, including cold-repressed exons (CREs), non-temperature sensitive exons (NTs) and cold-induced exons (CIEs) based on their percent spliced-in (PSI) at 35 °C and 39 °C (Fig. 1c). We found that in temperature-sensitive exons, G-content of sequences flanking the splice sites of CREs was significantly higher than that in CIEs and NTs. Such differences were not observed for sequences flanking splice sites of upstream and downstream constitutive exons, making a higher G-content specific for cold-repressed cassette exons (Supplementary Fig. 1a). Thus, these G-rich sequences may contribute to a mechanism controlling temperature-sensitive alternative splicing.

To further validate these findings in a different cell culture model, we performed RNA-Seq in Hela cells at 32, 37, and 40 °C (Supplementary Fig. 1b). In these RNA-Seq data, we identified 2093 CREs and 841 CIEs upon cold shock (32 °C vs. 37 °C) (Fig. 1d). We again observed a significantly higher G-content in sequences flanking the splice sites of CREs compared with that of CIEs and NTs (Fig. 1e). However, for those exons with splicing changes upon heat shock (40 °C vs. 37 °C), the enrichment was not observed (Supplementary Fig. 1c), suggesting that G-rich sequences may preferentially function at lower temperatures. We then asked whether these G-rich sequences could form rG4s by searching motifs from a previous study[27] and indeed observed a strong enrichment of rG4 motifs in sequences flanking the 5′-splice sites of CREs compared to that from NTs and CIEs (Fig. 1f). To confirm the presence of rG4s close to 5′-splice sites of CREs and to address the position of rG4s with respect to splice sites at single nucleotide resolution, we scanned sequences flanking each cassette exon with a 25 nt

window using G4Hunter[28]. The scores from G4Hunter scanning were higher for CREs than for CIEs and NTs, with the G4 score of 5′-splice sites of CIE being close to 0, indicating no rG4 in this region of CIEs (Fig. 1g). To further validate our in silico predictions, we analyzed RNA G-quadruplex sequencing (rG4-seq) data[29], which couples rG4-mediated reverse transcriptase stalling with high-throughput sequencing to map transcriptome-wide rG4s in Hela cells treated with the rG4 stabilizer PDS. PDS treatment leads to an enrichment of rG4-seq peaks around splice sites (Supplementary Fig. 1d), with a strong enrichment of rG4s flanking 3′ and 5′ splice sites of CREs (Fig. 1h, i).

Taken together, our analysis demonstrates that rG4 motifs are enriched around splice sites of CREs and are depleted from splice sites of CIEs. These results point to a possible role of rG4s in dynamically controlling the accessibility of splice sites to repress exon inclusion mainly at low temperatures, thereby serving as widespread regulators of temperature-dependent alternative splicing.

### Cold-repressed exons containing rG4s respond to G4 stabilizers

To address a global role of rG4s in controlling temperature-dependent alternative splicing, we correlated the predicted G4 scores with the changed exon inclusion levels (delta PSI) upon cold or heat shock treatment in HEK293T or Hela cells. Interestingly, we observed that the predicted G4 score around 5′-splice sites of temperature-sensitive exons correlated with increased exon exclusion in cold in both Hela and HEK293T cells and found the opposite, slightly weaker correlations upon heat shock (Fig. 2a). This result suggests a general correlation between folding probability (predicted G4 score) of rG4s around splice sites and the exclusion of the exon at low temperatures. To further test this, we first overlapped the identified CREs in Hela and HEK293T to obtain a high-confidence set of exons and to exclude exons with cell type-specific regulation, as splicing regulation directly through RNA structures could act independently of the *trans*-acting environment. In total, 1065 common CREs were identified in Hela and HEK293T cells and 380 (35.7%) of them contained predicted rG4s (score > 1) around their splice sites (Fig. 2b and Supplementary Data 1), pointing to a general, cell-type independent splice-regulatory function of rG4s. Next, we selected three candidate genes containing CREs with predicted rG4s around their splice sites, namely CDK4, IQSEC1 and FKBP15 (Fig. 2c). These genes have distinct functions in controlling cell cycle[30], phosphoinositide metabolism[31], and cytoskeletal organization[32], respectively. Sequence alignments showed that the G4 motifs surrounding these CREs are conserved cross multiple mammals, indicating their importance across evolution (Supplementary Fig. 2a, d, g). We then validated temperature sensitivity of splicing of these CREs and further investigated the impact of stabilizing rG4s on exon inclusion. Upon treating HEK293T and Hela cells with the widely used G4 stabilizer PDS[18], we measured inclusion levels of these CREs at different temperatures (33, 35, 37, and 40 °C). This analysis revealed a strong sensitivity of the CREs to the presence of PDS, which led to exon exclusion at higher temperature in both cell lines (Fig. 2d, e and Supplementary Fig. 2b, c, e, f, h, i, j), thus providing a first indication that rG4s may dynamically control temperature-regulated alternative splicing.

We also used two additional G4 ligands (Phen-DC3 and NMM) to address ligand-limited effects in HEK293 cells and tested several CREs (CDK4, IQSEC1, and RBM3). Phen-DC3 treatment significantly reduced the inclusion of these exons, whereas NMM had no effect, which is possibly due to differential responses of various stabilizers to different rG4 structures (Supplementary Fig. 3a–h).

### Identification of G-rich elements responsible for temperature-dependent RBM3 splicing

We next aimed to further substantiate the connection between G-rich elements and temperature-sensitive alternative splicing.

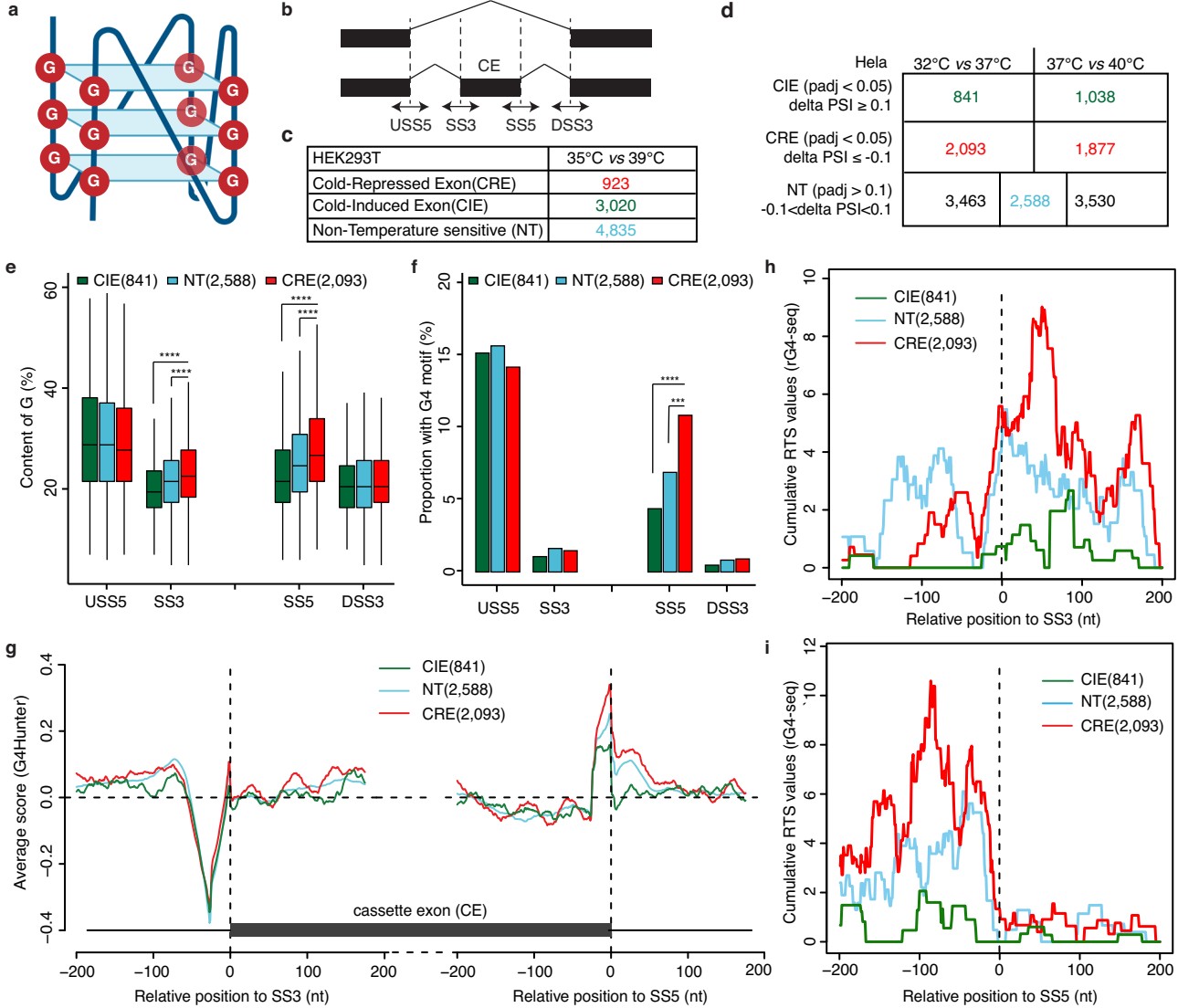

**Fig. 1 | G4 motifs are enriched around splice sites of cold-repressed exons (CREs). a** Schematic representation of a parallel G4 structure. It was generated with BioRender. **b** Schematic depicting the analyzed sequence in four regions around splice sites (SS) of each cassette exon (CE). USS5: upstream exon 5′ splice site, SS3: 3′ splice site, SS5: 5′ splice site, DSS3: downstream exon 3′ splice site. G-content and G4 motifs were quantified in 50 bp (−25 to +25) sequences flanking the respective splice sites. **c** The number of cold-repressed exons (CREs), cold-induced exons (CIEs), and non-temperature-sensitive exons (NTs) in HEK293T cells at 35 °C vs. 39 °C. CREs were defined as exons with ΔPSI ≤ −0.1 (35 °C–39 °C). CIEs were defined as ΔPSI ≥ 0.1 (35 °C–39 °C). Significance was assessed by a two-sided likelihood-ratio test with Benjamini–Hochberg-adjusted p-value < 0.05 between 35 °C and 39 °C. **d** The number of CIEs, CREs, and NTs in Hela cells in two comparisons (32 °C vs. 37 °C, 37 °C vs. 40 °C). The final NTs for further analysis were selected as the intersection of NT in these two comparisons (blue). **e** G-content in sequences within four regions (see **b**) of CREs, CIEs, and NTs in Hela cells (32 °C vs. 37 °C). The sample number of CIE (n = 841), NT (n = 2588), and CRE (n = 2093) were indicated in the figure. The box displays the interquartile range (IQR) with the median line. Whiskers extend to the most extreme data points within 1.5 × IQR of the box. Significance was estimated by two-sided Wilcoxon test (SS3: CRE vs. NT: p < 1.08e-13, CRE vs. CIE: p < 2.43e-18; SS5: CRE vs. NT: p < 3.35e-28, CRE vs. CIE: p < 1.99e-37). **f** Proportion of exons with G4 motifs[27] within four regions (see **b**) of CREs, CIEs, and NTs in Hela cells (32 °C vs. 37 °C). Significance was estimated by hypergeometric test (SS5: CRE vs. NT: p < 2.08e-5; CRE vs. CIE: p < 7.87e-8). **g** The average G4 scores predicted by G4Hunter around splice sites in CREs, CIEs, and NTs. 200 bp sequences flanking splice sites with a 25 bp window were searched starting at each base. **h**, **i** Cumulative Reverse Transcriptase Stalling (RTS) values based on rG4-seq[39] at the 3′ splice site (SS3, **h**) or the 5′ splice site (SS5, **i**) of alternative exons in CREs, CIEs, and NTs.

We focused on the cold-inducible and neuroprotective protein RBM3[10,33], which is controlled by temperature-regulated alternative splicing of a poison exon (exon 3a) leading to heat-induced NMD[11,34]. In RNA-Seq data, RBM3's exon 3a inclusion level remained consistently below 0.1 at or below 37 °C, while there was a significant increase observed at 39 °C or 40 °C in both HEK293T and Hela cells (Fig. 2f). In both hRBM3 and mRBM3 minigenes[11,34] (Fig. 2g), the splicing of exon 3a maintained temperature sensitivity, and the G4 ligand PDS[18] almost fully blocked exon 3a inclusion at all tested temperatures (Fig. 2h, i). Furthermore, we identified putative rG4 motifs around the 3′ and 5′-splice sites of this cold-repressed poison exon (Supplementary Fig. 4a, b). To explore the impact of G-rich elements near splice sites of RBM3 exon 3a, we introduced mutations to each G-rich element in five regions: two around the splice sites (R1 and R3), one within exon 3a (R2), and two in the downstream intron (R4 and R5) (Supplementary Fig. 4a, b). Interestingly, mutations or deletions of G-rich elements in R1 and R3 increased exon 3a inclusion in HEK293T (Supplementary Fig. 4c) and mouse N2a cells (Supplementary Fig. 4d). Using MaxEntScan[35], we further excluded the possibility that the effect of the mutations in R1 in

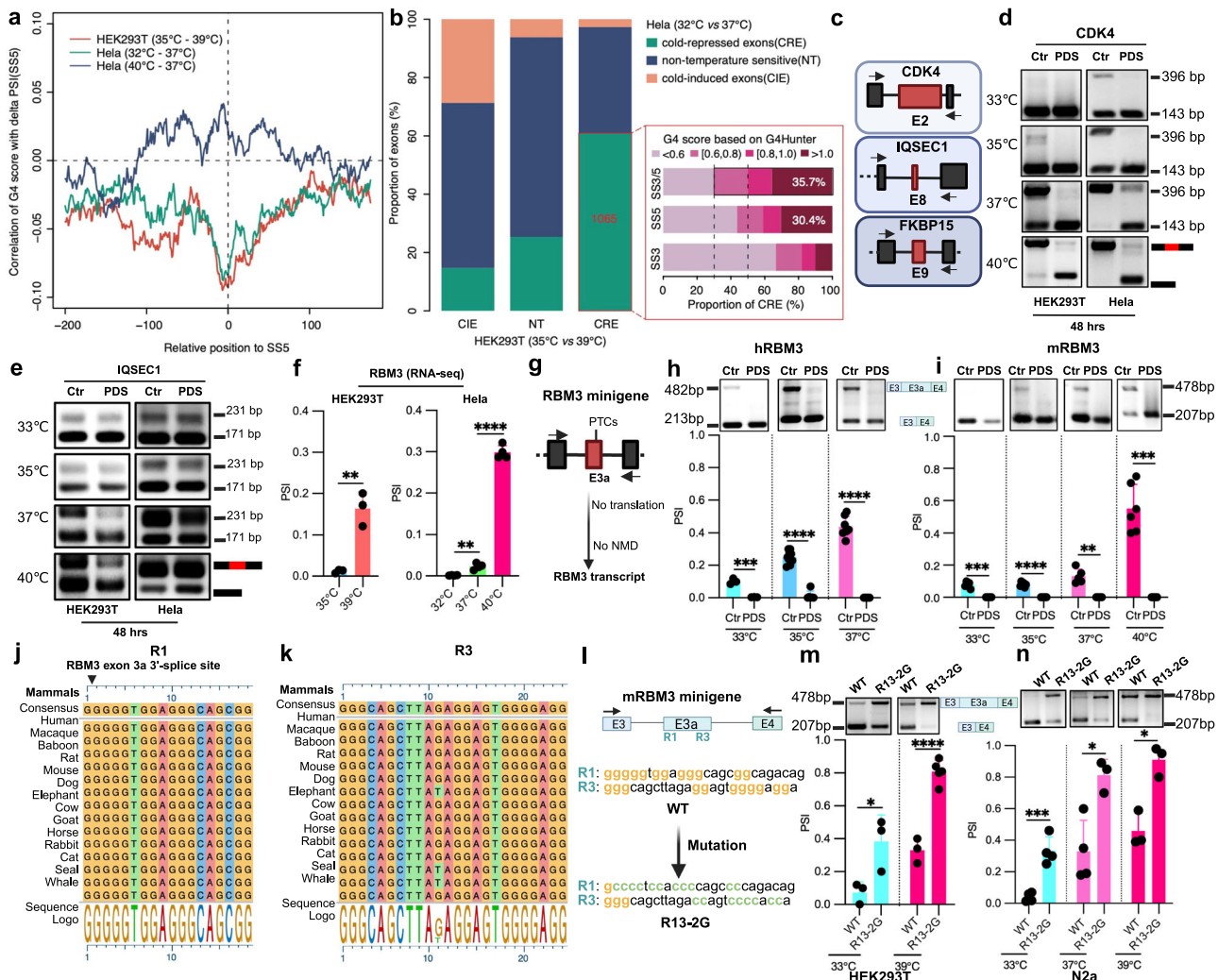

**Fig. 2 | G4 stabilizers suppress the inclusion of cold-repressed exons.**
**a** Correlation of 5′-splice site G4 scores with delta PSI values after cold shock
(HEK293T (35 °C–39 °C) and Hela (32 °C–37 °C)) or heat shock (Hela (40 °C–37 °C)).
For each base surrounding the splice site, the G4 score is calculated using a 25 bp
sliding window and then correlated with PSI. **b** Intersection of CIEs, NTs, and CREs
comparing HEK293T and Hela cells (left panel), and proportions of the shared CREs
containing sequences with different G4 scores (right panel). SS3/5 shows the
maximum G4 scores for either the SS3 or the SS5 for each exon. CIE cold-induced
exon, NT non-temperature sensitive exon, CRE cold-repressed exon. **c** Schematic
depicting the position of CREs (red) in CDK4, IQSEC1, and FKBP15. Arrows indicate
RT-PCR primers. It was generated with BioRender. **d**, **e** CRE inclusion level of CDK4
(**d**) and IQSEC1 (**e**) treated with DMSO or PDS at different temperatures in Hela
(right) and HEK293T (left) cells. A representative gel image is shown. PCR products
and sizes are indicated on the right. See Supplementary Fig. 2b, c (CDK4) and
Supplementary Fig. 2e, f (IQSEC1) for quantifications. **f** PSI of RBM3 exon 3a in
HEK293T and Hela RNA-Seq datasets from different temperatures. (see Fig. 1 and
bioinformatics "Method"). Data represent n = 3 (HEK293T) and n = 4 (HeLa) inde-
pendent biological replicates, with each data point corresponding to an individual
replicate sequencing dataset. Statistical significance was determined using an
unpaired two-tailed t-test. HEK293T: p = 0.0028 (35 °C vs. 39 °C); Hela: p = 0.0011
(32 °C vs. 37 °C), p < 0.0001 (37 °C vs. 40 °C). **g** Schematic of the RBM3 minigene
designed to prevent translation to allow analysis of exon inclusion independent of

NMD. PTCs: premature termination codons. This was generated with BioRender.
**h**, **i** Exon 3a inclusion in the hRBM3 (**h**) and mRBM3 minigene (**i**) after G4 stabilizer
PDS or control treatment in HEK293T cells. Upper gels depict representative
minigene-specific RT-PCR results, and the lower part shows quantified results ((**h**)
hRBM3: n = 3 for 33 °C, n = 9 for 35 °C (Ctr), n = 7 for 37 °C (Ctr) and n = 6 for PDS
(35 °C and 37 °C) ; (**i**) mRBM3: n = 6 for 35 °C and 40 °C, n = 5 for 33 °C and 37 °C).
**h** (hRBM3): p = 0.0007 (33 °C), p < 0.0001 (35 °C), p < 0.0001 (37 °C); **i** (mRBM3):
p = 0.0005 (33 °C), p < 0.0001 (35 °C), p = 0.0014 (37 °C), p = 0.0004 (40 °C).
**j**, **k** Sequence alignment of R1 and R3 across multiple mammals illustrated by a
derived consensus sequence, the aligned sequences, and a sequence logo repre-
sentation (see bioinformatics method). **l** Schematic illustrating the position of the
G-rich elements R1 and R3 in the mRBM3 minigene and the sequence of the R13-2G
mutant. It was generated with BioRender. **m**, **n** Splicing level of RBM3 exon 3a in WT
and R13-2G mutant at different temperatures in both HEK293T (**m**) and N2a cells
(**n**). Upper gels depict representative RT-PCR results, and the lower portion shows
quantified results. In HEK293T (**m**), n = 5 (R13-2G at 39 °C) and n = 3 (others); in N2a
cells (**n**), n = 4 (33 °C and WT at 37 °C) and n = 3 (others). **m** (HEK293T): p = 0.0362
(33 °C), p < 0.0001 (39 °C); **n** (N2a): p = 0.0008 (33 °C), p = 0.0121 (37 °C),
p = 0.0063 (39 °C). In this figure, individual data points are shown with mean +/−
SD. Statistical analysis was performed using unpaired two-tailed t-test. * denotes
p ≤ 0.05, ** denotes p ≤ 0.01, ***p ≤ 0.001 and **** denotes p ≤ 0.0001. Source data
are provided as a Source Data file.

promoting exon 3a inclusion was caused by increasing 3′-splice site
strength (Supplementary Fig. 4e).

In conclusion, the mutagenesis assay supports the hypothesis that
G-rich elements in regions R1 and R3 contribute to RBM3 exon 3a
alternative splicing through masking the closely adjacent splice sites or
by recruiting splice-regulatory proteins. The four stretches of

consecutive G residues in both R1 and R3, that potentially form
rG4 structures, are evolutionarily conserved in mammals (Fig. 2j, k),
indicating their importance in regulating RBM3 expression. An R13-2G
double mutant (Fig. 2l) increased isoforms containing exon 3a in both
HEK293T cells and N2a cells with a stronger effect than each single
mutant (Fig. 2m, n), again confirming that these G-rich sequences near

splice sites repress exon 3a inclusion. Further analysis using the RBM3 R13-2G minigene shows that it loses the responsiveness to G4 ligands, confirming that they act through the predicted G4 structures (Supplementary Fig. 4f, g; compared to Supplementary Fig. 3g, h).

To further validate the temperature-sensitive repressive effect of the G-rich elements R1 and R3 on alternative splicing, we introduced the R1 and/or R3 motifs into an unrelated minigene (MINX or a fluorescent reporter) and compared the effect on alternative splicing of the WT motif with its mutant or control counterparts. This approach enabled us to determine whether the G-rich motifs confer temperature-sensitive alternative splicing in an unrelated molecular setting in a cellular context. In the MINX minigene assay, we incorporated a segment of the RBM3 3a exon, including either the WT or mutated G-rich element, at the 3′ splice site of the MINX gene (Supplementary Fig. 5a). The WT sequence exhibited increased exon skipping compared to the mutant at lower temperature, with reduced skipping at elevated temperatures (Supplementary Fig. 5b, c), supporting the role of this G-rich R1 element as thermosensor in mammalian cells to regulate alternative splicing. Furthermore, we used a fluorescent minigene reporter system[36] (Supplementary Fig. 5d), where inclusion of a cassette exon disrupts the DsRed reading frame, preventing its expression while permitting GFP translation, whereas exon skipping maintains the DsRed frame, with a stop codon terminating translation before GFP is translated. The introduction of R1 and/or R3 near cassette exon splice sites significantly increased DsRed signal while decreasing GFP signal at both 37 °C and 39 °C, leading to a higher DsRed/GFP ratio (Supplementary Fig. 5e, f), indicative of increased exon skipping caused by R1 and/or R3. Notably, the inhibitory effect was reduced at 39 °C, highlighting the temperature-dependent characteristic of G-rich element-mediated splicing regulation. Importantly, this minigene exhibits temperature-responsiveness only in the presence of G-rich motifs near splice sites, supporting the hypothesis that these structures function as temperature sensors to modulate alternative splicing.

## RNA G-quadruplex formation in vitro revealed by biophysical assays

Our results demonstrate that G-rich elements surrounding RBM3 exon 3a regulate its inclusion, likely by limiting splice site accessibility at lower temperatures. To provide direct evidence that these sequences form rG4 structures and exhibit temperature sensitivity, we used an RNA oligonucleotide corresponding to the R1 sequence (rG4-R1) and a mutant version (R1 mutant), in which key G residues were replaced with Cs (Fig. 3a). Circular dichroism (CD) spectroscopy, a standard method for detecting rG4s[37] (Fig. 3b), revealed a strong positive peak at 265 nm and a negative peak at 240 nm for R1 RNA, characteristic of a parallel rG4 structure[37]. This signal was dependent on KCl[20], an endogenous G4 stabilizer, and absent in the R1 mutant, confirming rG4 formation in vitro (Fig. 3c). Further analysis showed that at low potassium concentrations (0.1 mM), rG4 formation was temperature-sensitive, with reversible signal intensity changes when heated to 40 °C and subsequently cooled to 33 °C (Fig. 3d). In contrast, at high potassium concentrations (50 mM), temperature sensitivity was reduced (Fig. 3e, f).

To further validate rG4 formation, we recorded 1D ¹H NMR spectra of rG4-R1 RNA under varying temperatures and potassium concentrations. The imino proton region (10–16 ppm) is indicative of RNA secondary structures, with Watson-Crick base-pairing signals appearing at 12–15 ppm and rG4-specific signals in a narrow range of 10.5–12.2 ppm[38]. In the absence of potassium, no imino signals were detected, indicating the lack of stable RNA structures. Upon potassium addition, well-dispersed imino signals at 27 °C confirmed rG4 formation. Temperature-dependent NMR analysis revealed that rG4 signals weakened between 37 °C and 42 °C (Fig. 3g), suggesting partial rG4 destabilization, and were nearly undetectable at 47 °C. In the aromatic

proton region (7.4–8.7 ppm), signals for unstructured RNA were absent at 27 °C but increased significantly at 37, 42, and 47 °C (Fig. 3h), indicating progressive rG4 unfolding at higher temperatures. Notably, the potassium concentrations used in vitro in both CD and NMR do not directly reflect cellular conditions, where additional factors contribute to rG4 stability. For example, RNA helicases (e.g., DHX36[39]) and RNA-binding proteins (e.g., hnRNPF[40]) can dissolve rG4s, which can make rG4s less stable in the cellular context even at the physiological potassium concentration of 140–150 mM[41].

Together, these results strongly support the formation of a temperature-responsive rG4 structure that dynamically responds to changes in the physiological temperature range. These findings together establish rG4s as mammalian RNA thermometers, capable of regulating temperature-dependent alternative splicing.

## RNA G-quadruplex formation in living cells

To further investigate the folding status of rG4 in living cells at different temperatures, we performed G4-specific immunostaining[42–44] and SHAPE-MaP probing[45,46] to directly assess rG4 folding dynamics. Cell exposure to low temperature (33 °C) significantly increased the G4-specific signal (Fig. 4a, b and Supplementary Fig. 6a–d), which was largely abolished by benzonase (a nuclease) but only partially reduced by DNase treatment, which affected mainly nuclear staining. This supports the notion that rG4 folding is promoted at lower temperatures in living cells and RNA, detected mainly in the cytoplasm, is capable of forming rG4. Furthermore, to check whether the perturbation of intracellular potassium concentration will influence the global G4 signal, we use voltage-gated potassium channel blocker (AFP, Amifampridine) to increase intracellular potassium in the glutamate-depolarized/-excited HT22 cells[47], and found AFP also significantly increased the global G4 signal in these cells (Supplementary Fig. 6e, f), indicating that increasing intracellular potassium promotes rG4 formation.

To further substantiate the role of rG4s as physiological thermosensors to modulate alternative splicing in mammalian cells, we performed SHAPE-MaP probing[45,46] (Fig. 4c, d). In this method, NAI (2-methylnicotinic acid imidazolide) preferentially modifies the 2′-hydroxyl group of the ribose sugar in flexible regions of RNA, leading to higher modification of the loop nucleotides connecting the G-tetrads, of bulged residues within the G4 structure and of unstructured regions adjacent to the G4 core, while regions with stable base pairing and the G-tetrad stacking region will show reduced modification[26,48]. The NAI-induced modification will lead to mutations or indels in the cDNA during reverse transcription with Mn²⁺ using Superscript II reverse transcriptase[45,46], which can be quantified by the TIDE software[49]. Our results demonstrated that treatment with the G4 ligands (PDS and Phen-DC3) in living cells significantly increased the indel percentage surrounding G-rich motifs near splice sites in several cold-repressed exons (RBM3, CDK4, and FKBP15), suggesting that these motifs can form rG4 structures in live cells. More importantly, the indel percentage was significantly increased at low temperatures (33 °C) (Fig. 4e, f, Supplementary Fig. 7a, b, and Supplementary Fig. 8a–d), supporting that these G-rich elements form more rG4s under cold conditions. Taken together, these findings provide strong evidence that rG4s function as physiological potassium and thermosensors that can then modulate downstream processing events.

## RBM3 rG4 elements control RBM3 levels in response to G4 stabilizers

After demonstrating that the G-rich sequence around splice sites of RBM3 exon 3a can form rG4 structures in vitro and in vivo, we aimed to confirm their relevance for RBM3 exon 3a splicing in cells. Thus, we turned to the endogenous rG4 stabilizer potassium (K⁺) and tested the impact of KCl on the mRBM3 minigene and the rG4 double mutant (R13-2G) in HEK293T and N2a cells. In the WT mRBM3 minigene, KCl

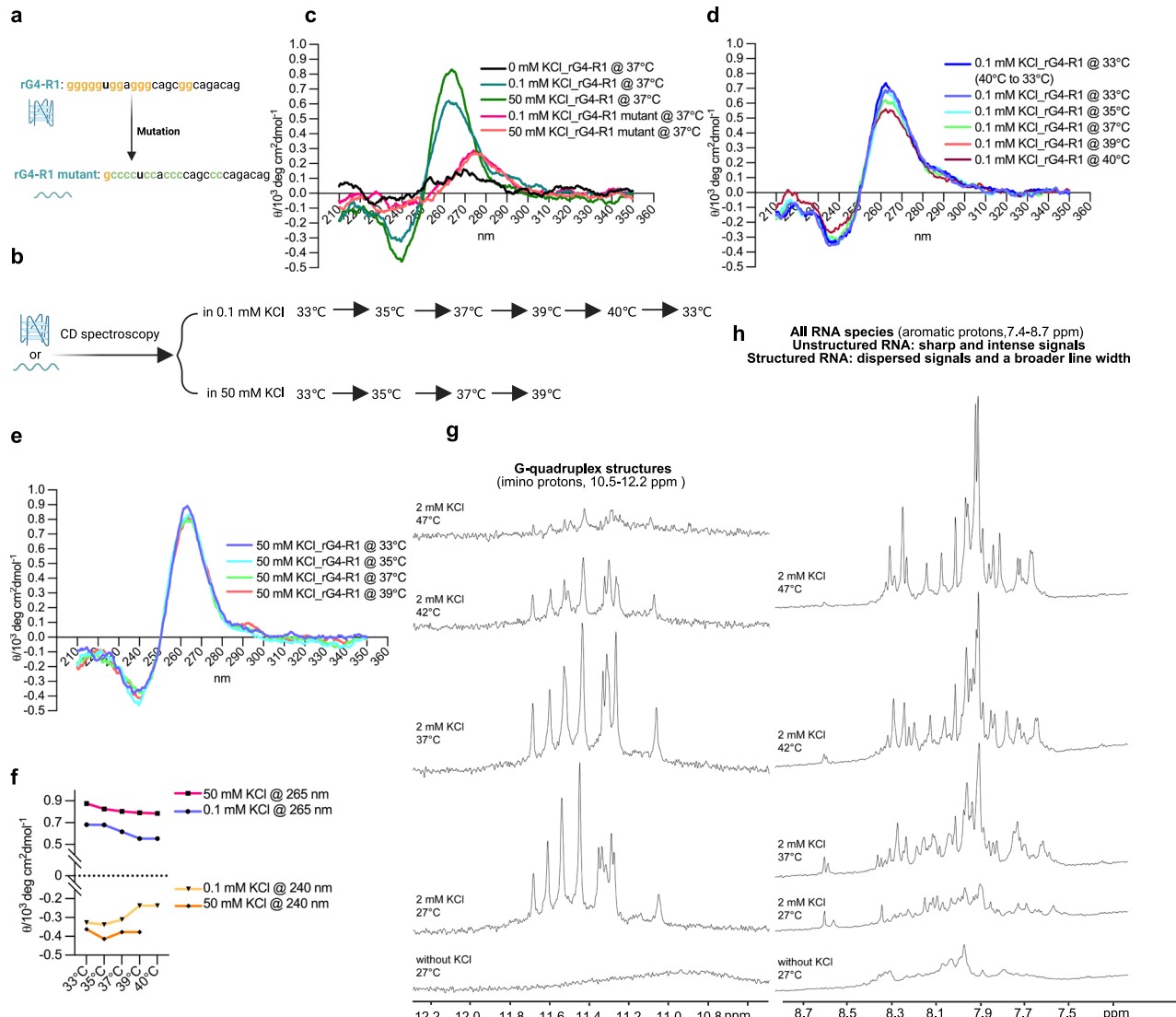

**Fig. 3 | Biophysical assays demonstrate temperature and potassium dependent G-quadruplex formation of RBM3 R1 in vitro. a** Sequences of synthesized WT and mutant rG4-R1 RNAs. R1 corresponds to the sequence near the 3′ splice site of RBM3 eoxn 3a (Fig. 2j). **b** A schematic of biophysical CD spectroscopy assays. The RNA samples were heated to 95 °C for 5 min and then cooled to room temperature overnight for renaturation. Subsequently, the samples were scanned from 220 to 310 nm at different temperatures. After setting the CD spectrometer to the desired measurement temperature, RNA samples were incubated at that temperature for 30 min to ensure thermal equilibrium and structural stabilization. CD spectra were then recorded using a response time of 0.5 s per nm. Each spectrum represents the average of three consecutive scans collected under these conditions. It was generated with BioRender. **c** Circular dichroism (CD) spectrum signal of rG4-R1 and its mutant in different potassium concentrations at 37 °C. *X*-axis: the wavelength of light; *y*-axis: the magnitude of the CD signal. Signal peak at 265 nm and 240 nm: parallel rG4 structure. **d**, **e** Normalized CD signal of rG4-R1 at different temperatures in low KCl concentration (**d**) and high KCl condition (**e**). $[\theta] = \theta_{obs} \times 100/(c \times l)$, where $[\theta]$ = molar ellipticity in deg·cm²·dmol⁻¹; $\theta_{obs}$ = observed ellipticity in milli-degrees (mdeg); $c$ = molar concentration of the sample in mol/L; $l$ = path length in cm; Multiply by 100 to convert mdeg to deg and account for units. **f** Summary of CD signal peaks at 265 nm (top) and 240 nm (bottom) of rG4-R1 at various temperatures under both high and low concentrations of KCl. **g**, **h** 1D ¹H NMR spectrum showing the imino region (**g**, peaks indicating structured rG4s) and aromatic protons (**h**, as a measure for unstructured RNA, see text for details), confirming that the rG4-R1 sequence forms a temperature-sensitive rG4 in vitro. Source data are provided as a Source Data file.

treatment significantly decreased RBM3 exon 3a inclusion at high temperatures, while it had no effect at low temperatures (Fig. 5a–d). This suggests that higher stability of the rG4 structures at lower temperatures leads to reduced dependence on potassium, and at high temperature, potassium stabilizes the rG4, leading to more RBM3 exon3a skipping. In contrast, exon 3a inclusion in the double mutant (R13-2G) did not significantly change upon KCl treatment at both low and high temperatures (Fig. 5a–d). Consistently, KCl promoted exon 3a skipping in the WT mRBM3 minigene, while it had no effect on the R13-2G mutant minigene in HT22 cells at normal physiological temperature 37 °C (Fig. 5e, f). In the polarized/glutamate-activated HT22 cells, which mimics pathological brain injury condition[47], both 4-AP

and AFP significantly, reduced exon 3a inclusion in the WT, but not in the R13-2G minigene (Fig. 5g, h). These results together strongly suggest that rG4s in RBM3 exon 3a are KCl-responsive and temperature-dependent RNA elements that, by masking splice sites, control exon inclusion in cells.

**Elevation of endogenous potassium enhances RBM3 expression**
Since RBM3 exon 3a inclusion triggers nonsense-mediated decay (NMD), promoting exon 3a skipping by stabilizing rG4 near its splice sites using potassium can enhance endogenous RBM3 expression. We therefore used KCl to stabilize rG4s in RBM3 exon 3a and investigated the impact on endogenous RBM3 expression. In line with our

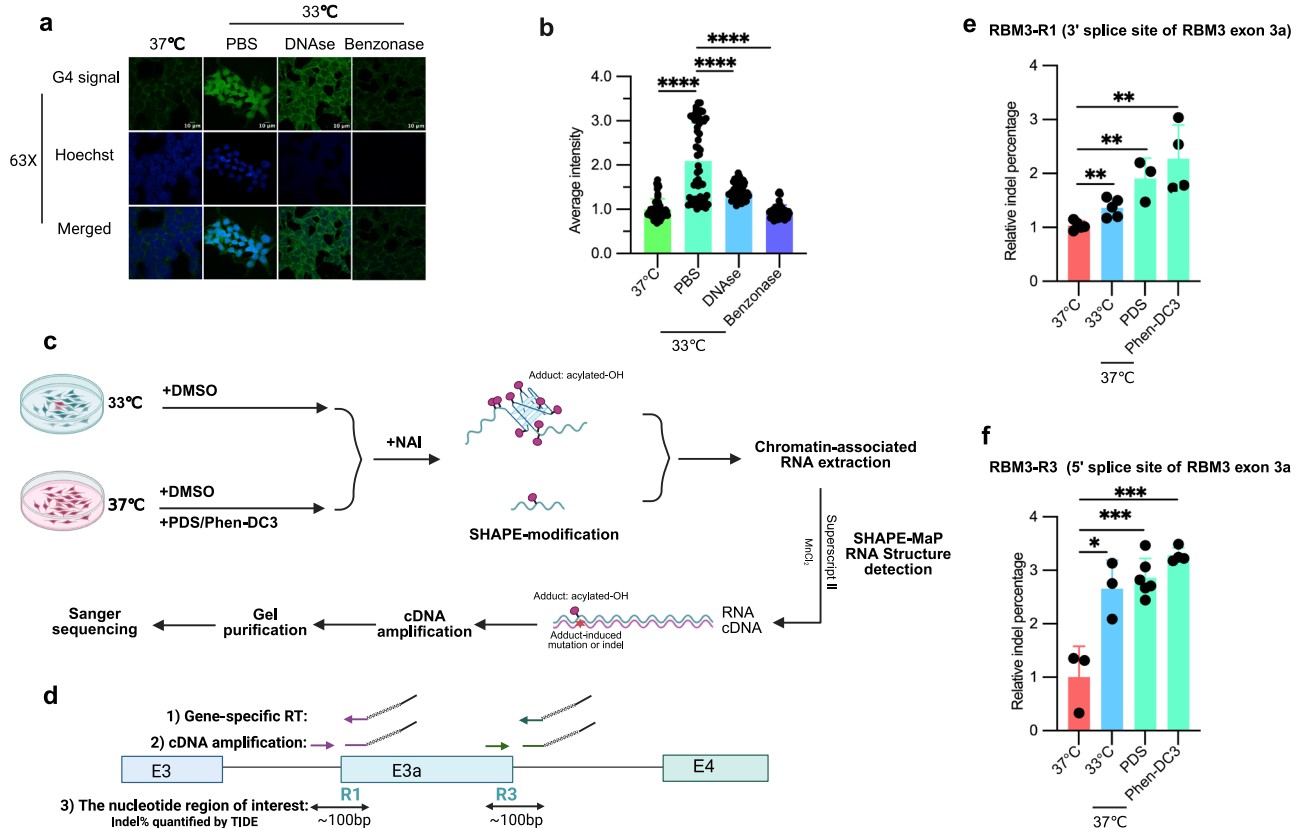

**Fig. 4 | G4-specific immunostaining and SHAPE-MaP show temperature and potassium dependent G-quadruplex formation in live cells. a** Representative G4-specific immunostaining images of HEK293 cells at different treatments. Cells were treated at different temperatures, followed by immunostaining with G4-specific antibody (Anti-DNA/RNA G-quadruplex [BG4], Absolute Antibody, Ab00174-30.126)[42–44] and confocal imaging. DNAse is to digest DNA. Benzonase is to digest both RNA and DNA. **b** Quantification of (**a**), with Fiji (ImageJ2 V2.16.0/1.54g) software. Quantifications were based on three biological replicate images, with signal intensity measured from $n = 18$ cells for the PBS control and $n = 17$ cells for the treatment conditions per image. Each dot represents the mean G4 signal intensity overlapping with the Hoechst-stained nuclear region in a single cell. Each dot represents the average intensity of G4 signal overlapped with Hoechst signal position in a single cell. The scatter dot plots represent mean ± SD. $p < 0.0001$ (37 °C vs. PBS), $p < 0.0001$ (PBS vs. DNAse), $p < 0.0001$ (PBS vs. Benzonase). **c** Schematic of SHAPE-MaP probing method. HEK293 cells were treated with G4 stabilizers (PDS or Phen-DC3) or DMSO for 24 h or incubated at different temperatures for 48 h, and then cells were then treated with 50 mM NAI. To get the enriched pre-mRNA, Chromatin-associated RNA was extracted, reverse-transcribed in an Mn²⁺-containing buffer, followed by cDNA amplification, gel-purification, and

analysis via Sanger sequencing. It was generated with BioRender. **d** Schematic of the region of interest in SHAPE-MaP using RBM3 R1 and R3 as examples. Chromatin-associated RNA was extracted and reverse-transcribed into cDNA in an Mn²⁺-containing buffer using Superscript II with a gene-specific primer binding -50 bp downstream of the putative rG4. The cDNA was amplified with a high-fidelity polymerase and primers targeting -50 bp upstream and downstream of the putative rG4. It was generated with BioRender. **e, f** Quantified indel percentage of Sanger sequencing data from (Supplementary Fig. 7). In the **e** (RBM3-R1): data were obtained from $n = 5$ biological replicates for the 33 °C and 37 °C control treatments, $n = 3$ for PDS treatment, and $n = 4$ for Phen-DC3 treatment; **f** (RBM3-R3): $n = 3$ for the 33 °C and 37 °C control treatment, $n = 6$ for PDS, and $n = 4$ for Phen-DC3. The indel percentages were quantified using TIDE software within the 50 bp upstream and downstream regions of the putative rG4[49]. In the **e** (RBM3-R1): $p = 0.0057$ (37 °C vs. 33 °C), $p = 0.0021$ (37 °C vs. PDS), $p = 0.0031$ (37 °C vs. Phen-DC3); **f** (RBM3-R3): $p = 0.0214$ (37 °C vs. 33 °C), $p = 0.0005$ (37 °C vs. PDS), $p = 0.0005$ (37 °C vs. Phen-DC3). In this figure, the scatter dot plots represent mean ± SD. Statistical analysis was performed using two-tailed unpaired $t$-test. * denotes $p \leq 0.05$, ** denotes $p \leq 0.01$, ***$p \leq 0.001$ and **** denotes $p \leq 0.0001$. Source data are provided as a Source Data file.

hypothesis, KCl treatment increased RBM3 mRNA and protein expression at high (≥ 37 °C) but not at low (< 37 °C) temperatures in HEK293T and N2a cells (Fig. 5i–l). Collectively, these findings indicate that both low temperatures and potassium ions are capable of stabilizing rG4 structures, thereby increasing endogenous RBM3 expression.

## 4-AP-mediated neuronal protection against hemin-induced damage in an RBM3-dependent manner

As it has been well established that RBM3 safeguards neurons and ameliorates phenotypes in prion disease and hypoxic-ischemic brain injury mouse models by facilitating neuronal structural plasticity, preventing cell death, and promoting neurogenesis[10,33], we subsequently investigated whether stabilizing rG4 structures by increasing intracellular potassium concentration with 4-AP, a clinically approved compound, could provide the neuroprotective effects in a hemin-

induced hemorrhagic cell model[50]. In this model, hemin, a derivative of heme—an essential component of hemoglobin in red blood cells—mimics the oxidative stress and neuroinflammation of early brain injury in hemorrhagic stroke. We observed that hemin treatment led to a notable reduction in intracellular potassium concentration, whereas 4-AP treatment produced a dose-dependent increase of intracellular potassium in the hemin-induced cell culture model (Fig. 6a). We next examined the effect of 4-AP on RBM3 in the hemorrhagic model. We first measured the splicing levels of RBM3 exon 3a using three distinct primer pairs: two specifically targeting inclusive exon 3a junctions (primer pair 1 for 3′ splice site junction and pair 2 for 5′ splice site junction) (Fig. 6b) and the other one amplifying total RBM3 mRNA. It shows that hemin significantly prompted exon 3a inclusion, as evidenced by a higher ratio of exon 3a to RBM3 mRNA (Fig. 6c, d), ultimately resulting in an almost two-fold reduction of RBM3 expression (Fig. 6e), while 4-AP treatment significantly led to a gradual reduction

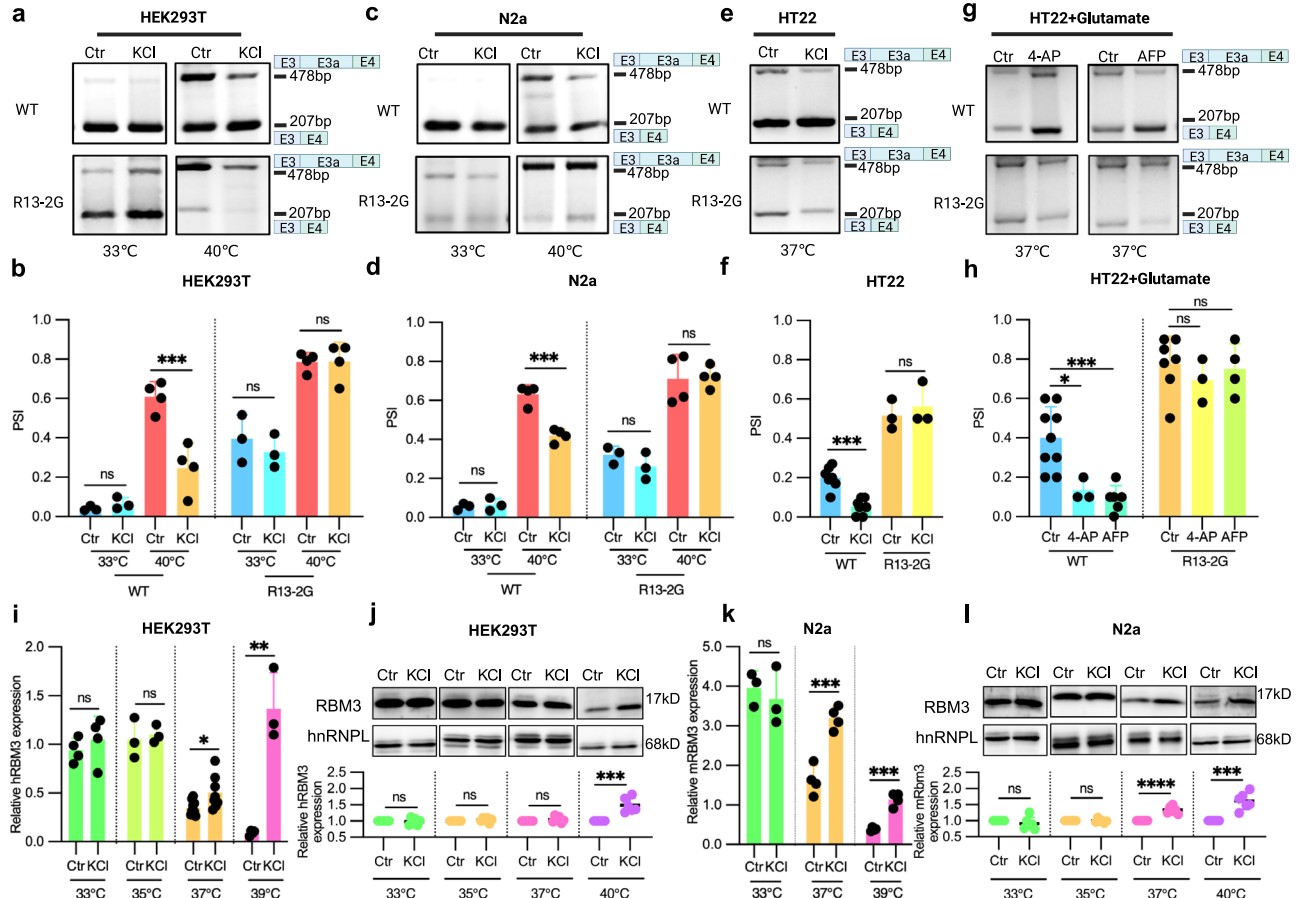

**Fig. 5 | RBM3 rG4 elements control RBM3 levels in response to G4 stabilizers.**
**a–d** Splicing level of RBM3 exon 3a in the WT and R13-2G double mutant mRBM3 minigene after 50 mM KCl and control treatment at 33 °C and 40 °C in both HEK293T (**a** and **b**) and N2a (**c** and **d**) cells. Data were obtained from $n = 3$ biological replicates for the 33 °C and $n = 4$ for 40 °C treatments. Gels in (**a** and **c**) depict representative minigene-specific RT-PCR results, and the lower part (**b** and **d**) shows quantified results of (**a** and **c**). **b** (HEK293T): $p = 0.0025$ (WT Ctr vs. KCl at 40 °C), **d** (N2a): $p = 0.0004$ (WT ctr vs. KCl at 40 °C). **e, f** Splicing level of RBM3 exon 3a after KCl treatment as in (**a–d**) in HT22 cells at 37 °C. Data were obtained from $n = 7$ biological replicates for the WT and $n = 3$ for R13-2G treatments. Gels in (**e**) depict representative minigene-specific RT-PCR results, and the lower part (**f**) shows quantified results of (**e**). **f** (HT22): $p = 0.0001$ (WT ctr vs. KCl at 40 °C). **g, h** Splicing level of RBM3 exon 3a after voltage-gated potassium channel blocker (4-AP and AFP) treatment in glutamate-stimulated HT22 cells at 37 °C. Data represent $n = 9$ for WT control, $n = 3$ for WT + 4-AP, $n = 6$ for WT + AFP, $n = 7$ for R13-2G control, $n = 3$ for R13-2G + 4-AP and $n = 4$ for R13-2G + AFP treatment. 4-AP: 4-Aminopyridine. AFP: amifampridine. 10 μM of 4-AP and AFP were used. Glutamate

is used to polarize and excite HT22 cells. **h** (HT22): $p = 0.0194$ (WT Ctr vs. 4-AP), $p = 0.0006$ (WT Ctr vs. AFP). **i–l** RBM3 expression after 50 mM KCl and control treatment at 33, 35, 37, 39 °C (for RNA level) or 40 °C (for protein level), observed for both mRNA and protein levels in HEK293T (**i** and **j**) and N2a (**k** and **l**) cells. **i** and **k** are derived from qPCR (in (**i**): $n = 4$ for 33 °C, $n = 3$ for 35 °C, $n = 9$ for 37 °C control, $n = 8$ for 37 °C + KCl, $n = 4$ for 39 °C control, and $n = 3$ for 39 °C + KCl treatment; in (**k**): $n = 3$ for 33 °C, $n = 4$ for others). **j** and **l** are WB results ($n = 6$ biological replicates). qPCR was normalized with hHPRT. Below the gels, quantifications using HNRNPL as loading control are shown (see "Method"). **i** (HEK293T): $p = 0.0224$ (Ctr vs. KCl at 37 °C), $p = 0.0009$ (Ctr vs. KCl at 39 °C); **j** (HEK293T): $p = 0.0003$ (Ctr vs. KCl at 40 °C); **k** (N2a): $p = 0.0005$ (Ctr vs. KCl at 37 °C), $p = 0.0001$(Ctr vs. KCl at 39 °C); **l** (N2a): $p < 0.0001$ (Ctr vs. KCl at 37 °C), $p = 0.0002$ (Ctr vs. KCl at 40 °C). In this figure, individual data points and mean ± SD are shown. Statistical analysis was performed using two-tailed unpaired $t$-test. ns denotes no significance, * denotes, $p \leq 0.05$, **$p \leq 0.01$ and *** denotes $p \leq 0.001$. Source data are provided as a Source Data file.

in the exon 3a to total RBM3 ratio (Fig. 6c, d) and an increase in RBM3 mRNA expression (Fig. 6e). Consequently, RBM3 protein expression was substantially increased following 4-AP treatment (Fig. 6f and Supplementary Fig. 9a, b), reaching a nearly five-fold elevation compared to hemin treatment alone (Fig. 6f). Phenotypically, the hemin-induced cell death rate decreased (Fig. 6g, h), from 25% to 5%, and cell viability increased significantly and substantially following 4-AP treatment (Fig. 6i). Knocking down RBM3 (Fig. 6j) with siRNA post-4-AP treatment abolished these phenotypic effects (Fig. 6j–m), underscoring the pivotal role of RBM3 in mediating the neuroprotection conferred by 4-AP. To further validate RBM3's protective role against hemin-induced neuronal damage, we overexpressed RBM3 in HT22 cells (Supplementary Fig. 9c, d). This led to an increase of cell viability from 60% to 80% (Supplementary Fig. 9e) and a notable decline in cell

death from 16% to 7% (Supplementary Fig. 9f, g) upon hemin treatment.

## Neuronal protection of 4-AP in a subarachnoid hemorrhage mouse model

To further address 4-AP's efficacy and potential for clinical applications, we subsequently evaluated the impact of 4-AP in vivo using a subarachnoid hemorrhage (SAH) mouse model[51,52]. The dosage of 4-AP in mice was based on previously established protocols to ensure effectiveness while minimizing significant side effects[53,54] (Fig. 7a). Given that the most impacted area in individuals with SAH is near the bleeding site, we directed our attention to the cerebral cortex surrounding the affected region (Fig. 6b). We first examined the RBM3 expression level after 4-AP administration. Intraperitoneal

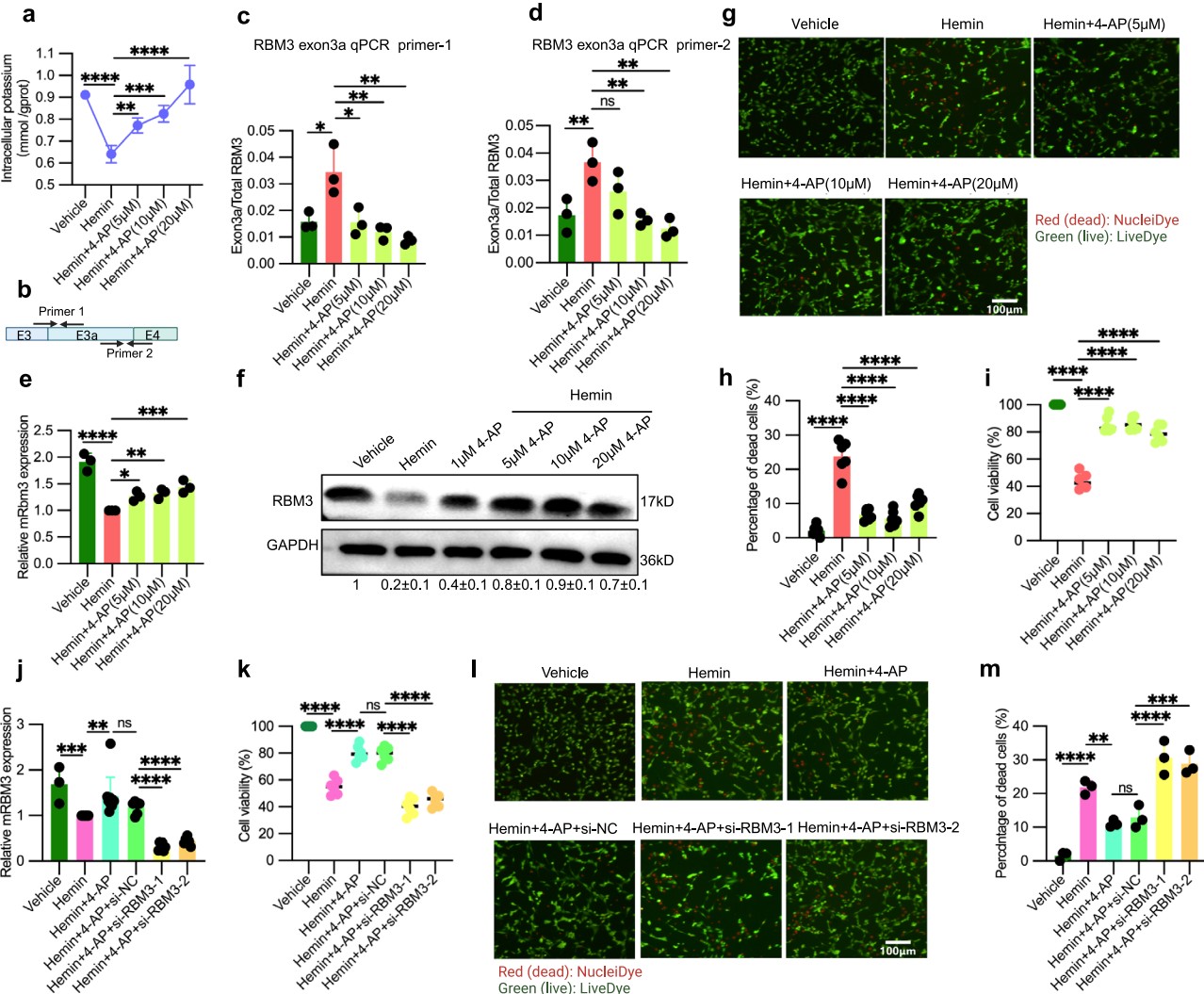

**Fig. 6 | 4-AP confers RBM3-mediated protection against neuronal damage in a hemin-induced hemorrhage HT22 cell model. a** Intracellular potassium levels after 4-AP and control treatment in hemin-exposed HT22 cells (see "Method"). Data represent $n = 4$ independent biological replicates (HT22 cells). The plot shows points connected by a line with error bars (mean ± SD). $p < 0.0001$ (Vehicle vs. Hemin), $p = 0.0065$ (Hemin vs. Hemin+4-AP (5 μM)), $p = 0.0002$ (Hemin vs. Hemin +4-AP (10 μM)), $p < 0.0001$ (Hemin vs. Hemin+4-AP (20 μM)). **b** Schematic of qPCR primers for RBM3 exon 3a. Primer pair 1 targets the upstream exon–exon 3a junction. Primer pair 2 targets the exon 3a–downstream exon junction. It was generated with BioRender. **c, d** Levels of RBM3 exon 3a in HT22 cells treated as in (**a**). Two pairs of qPCR primer (RBM3 exon3a qPCR primer-1 (**c**) and RBM3 exon3a qPCR primer-2 (**d**)) were used to quantify exon 3a inclusion, respectively. Data represent $n = 3$ independent biological replicates (HT22 cells). **c** $p = 0.0167$ (Vehicle vs. Hemin), $p = 0.0155$ (Hemin vs. Hemin+4-AP (5 μM)), $p = 0.0046$ (Hemin vs. Hemin+4-AP (10 μM)), $p = 0.0016$ (Hemin vs. Hemin+4-AP (20 μM)); **d** $p = 0.0078$ (Vehicle vs. Hemin), $p = 0.0044$ (Hemin vs. Hemin+4-AP (10 μM)), $p = 0.0014$ (Hemin vs. Hemin+4-AP (20 μM)). **e** qPCR analysis of RBM3 total mRNA expression after treatments as in (**a**). $n = 3$ independent biological replicates (HT22 cells). qPCR was normalized with mGAPDH. $p < 0.0001$ (Vehicle vs. Hemin), $p = 0.0280$ (Hemin vs. Hemin+4-AP (5 μM)), $p = 0.0091$ (Hemin vs. Hemin+4-AP (10 μM)), $p = 0.0008$ (Hemin vs. Hemin+4-AP (20 μM)). **f** Endogenous RBM3 protein expression in HT22 treatments as in (**a**), analyzed by Western blot. Below the gel, quantification (mean ± SD) using GAPDH as loading control is shown. Data represent $n = 3$ independent biological replicates (HT22 cells). $p < 0.0001$ in all the comparisons (Vehicle vs. Hemin, Hemin vs. Hemin+4-AP (5 μM), Hemin vs. Hemin+4-AP (10 μM), Hemin vs. Hemin+4-AP (20 μM)). **g, h** Cell death in HT22 cells treated as in (**a**). Representative images stained for dead/live cells are shown in (**g**), with quantification of % dead cells in (**h**) (see "Method"). Data represent $n = 3$ independent biological replicates (HT22 cells). Red fluorescence indicates dead cells stained with NucleiDye, while green fluorescence marks live cells stained with LiveDye. In (**h**), $p < 0.0001$ in all the comparisons (Hemin vs. Hemin+4-AP (5 μM), Hemin vs. Hemin+4-AP (10 μM), Hemin vs. Hemin+4-AP (20 μM)). **i** Cell viability was investigated in HT22 treated as in (**a**) by an CCK-8 assay and is plotted as % viable cells (see "Method", $n = 6$ independent biological replicates (HT22 cells)). $p < 0.0001$ in all the comparisons. **j** Relative RBM3 mRNA expression post-siRNA transfection upon 4-AP treatment in hemin-exposed HT22 cells. Two independent si-RNAs against RBM3 (RBM3-1 and RBM3-2) were used (see "Method"). Data represent $n = 3$ independent biological replicates for Vehicle, $n = 10$ for Hemin and Hemin + 4-AP, and $n = 6$ for other treatments (HT22 cells). $p = 0.0005$ (Vehicle vs. Hemin), $p = 0.0015$ (Hemin vs. Hemin+4-AP), $p < 0.0001$ (Hemin+4-AP+si-NC vs. Hemin+4-AP+si-RBM3-1), $p < 0.0001$ (Hemin+4-AP+si-NC vs. Hemin+4-AP+si-RBM3-2). **k** Cell viability in treatments as in (**j**), quantified by CCK-8 assay. Data represent $n = 6$ independent biological replicates (HT22 cells). $p < 0.0001$ in all the comparisons. **l, m** Percentage of dead cells in treatments as in (**j**). Representative staining for dead/live cells is provided in (**l**), with quantification in (**m**) (see "Method"). Data represent $n = 3$ independent biological replicates (HT22 cells). $p < 0.0001$ (Vehicle vs. Hemin), $p = 0.0048$ (Hemin vs. Hemin+4-AP), $p < 0.0001$ (Hemin+4-AP+si-NC vs. Hemin+4-AP+si-RBM3-1), $p = 0.0002$ (Hemin+4-AP+si-NC vs. Hemin+4-AP+si-RBM3-2). In this figure, the scatter dot plots are shown as the mean ± SD. Statistical analysis was conducted using ordinary one-way ANOVA followed by Dunnett's multiple comparisons test (**j**, **k**, and **m** Šídák's multiple comparisons test) with a single pooled variance. ns denotes no significance, * denotes $p \leq 0.05$, ** denotes $p \leq 0.01$, *** denotes $p \leq 0.001$ and **** denotes $p \leq 0.0001$. Source data are provided as a Source Data file.

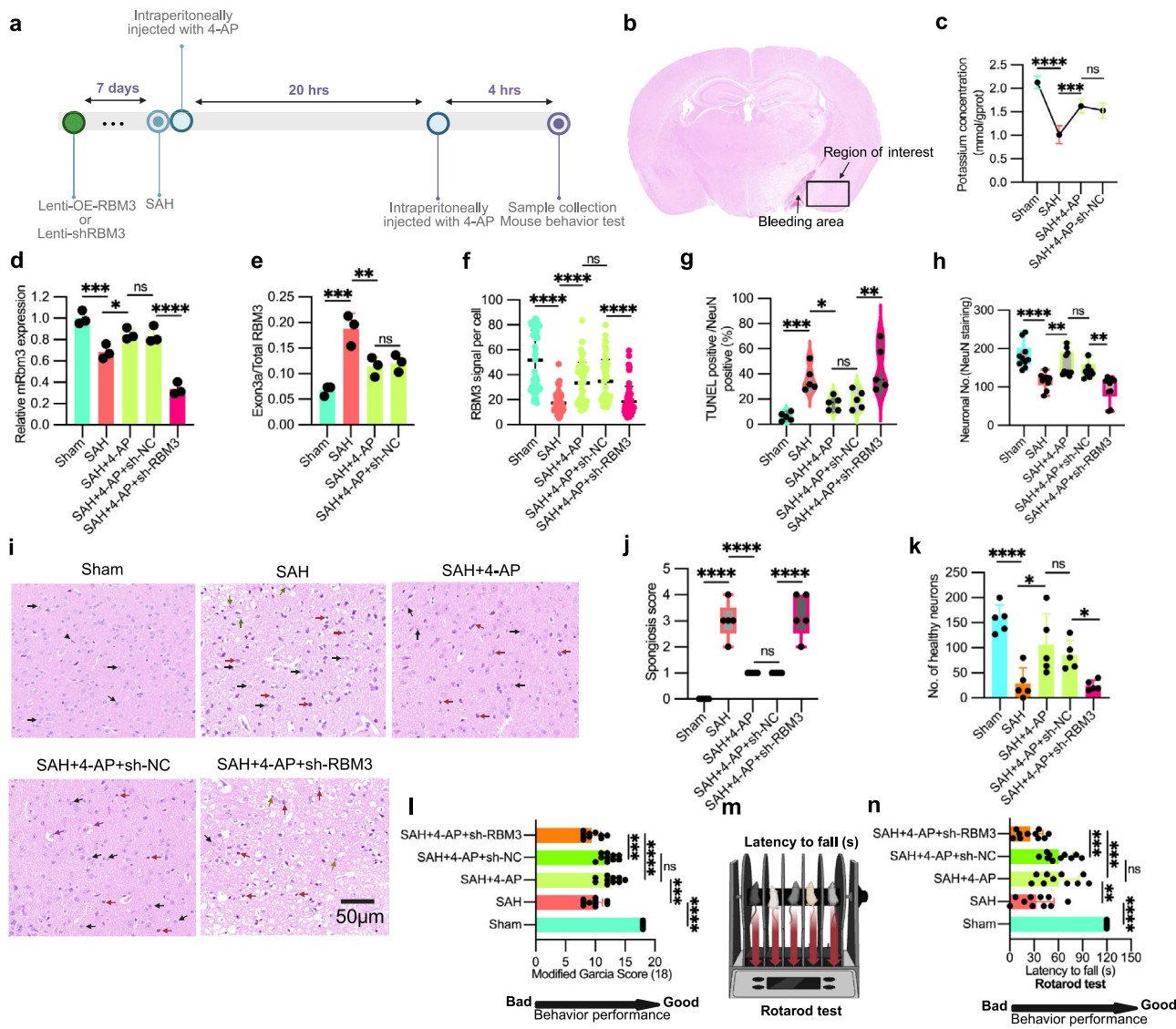

administration of 4-AP resulted in a significantly elevated intracellular potassium level in the cerebral cortex (Fig. 7c), upregulation of endogenous RBM3 mRNA expression (Fig. 7d) and decreased RBM3 exon 3a inclusion (Fig. 7e and Supplementary Fig. 10a, b). RBM3 protein expression was also augmented, as demonstrated by immunostaining (Fig. 7f and Supplementary Fig. 10c). Next, neuronal apoptosis, neuronal count, and cortical spongiosis were analyzed as key readouts to evaluate the effects of 4-AP in the SAH model. Notably, the increased RBM3 expression following 4-AP treatment correlated with decreased neuronal apoptosis (Fig. 7g) from ~30% to 15%, as indicated by TUNEL staining[55] (Supplementary Fig. 10d), and increased neuronal count (Fig. 7h) shown by neuronal nuclear protein (NeuN) staining[56] (Supplementary Fig. 10c, d). Hematoxylin and eosin (HE) staining also revealed that 4-AP treatment significantly ameliorated the severity of cortical spongiosis[57] resulting from SAH (Fig. 7i, j). 4-AP administration also prevented neuronal damage, which is apparent as eosinophilic necrotic neurons with aberrant morphology characterized by cell body shrinkage, darkly stained pyknotic nuclei, and intensely stained eosin, with a significant increase in neuronal count from around 20 to 100 (Fig. 7i, k). Further, Nissl staining[58] demonstrated increased cell counts with normal morphology following 4-AP treatment compared to the SAH control group (Supplementary Fig. 10e, f). Finally, since the SAH model induces hemorrhage near the circle of Willis, leading to acute neurological deficits due to subarachnoid blood accumulation and

secondary ischemia, we assessed motor function using the rotarod test, a standard measure of coordination and balance in rodents, and evaluated overall neurological deficits with the modified Garcia score. 4-AP treatment markedly improved mouse behavioral performance, manifested by increased modified Garcia scores[58,59] (Fig. 7l) and enhanced latency to fall in the Rotarod test (Fig. 7m, n).

To confirm whether the beneficial effects of 4-AP treatment in the SAH mouse model are dependent on increased RBM3 expression, we utilized an shRNA lentivirus to knock down RBM3 in vivo prior to SAH and 4-AP treatment (Fig. 7a–d, f and Supplementary Fig. 10c). We then evaluated whether RBM3 knockdown abolished the neuroprotection conferred by 4-AP using previously established neuronal readouts and behavioral assessments. Consequently, this resulted in increased neuronal apoptosis[55] (Fig. 7g and Supplementary Fig. 10d), reduced neuronal count (Fig. 7k), as evidenced by NeuN staining (Supplementary Fig. 10c, d), elevated spongiosis score (Fig. 7i, j), and decreased healthy neurons (Fig. 7k) shown by HE staining (Fig. 7i), decreased cells with normal morphology indicated by Nissl staining[58] (Supplementary Fig. 10e, f), compared to 4-AP treatment alone. Moreover, the improved mouse behavior elicited by 4-AP treatment was nullified by RBM3 knockdown, as evidenced by decreased modified Garcia scores[58,59] (Fig. 7l) and reduced latency to fall in the Rotarod test (Fig. 7m, n). Taken together, these results strongly support the role of RBM3 as a key mediator of the neuronal effects of 4-AP.

**Fig. 7 | 4-AP mitigates neuronal damage in a mouse model of subarachnoid hemorrhage (SAH). a** Timeline of in vivo mouse experiments. Lentivirus (OE-RBM3 or shRNA against RBM3) were injected into the left cerebral cortex 7 days before SAH onset. 4-AP was injected intraperitoneally immediately after SAH. Due to its short half-life, 4-AP was injected again after 20 h SAH. All the mouse samples were collected after 24 h SAH. It was generated with BioRender. **b** Representative mouse cerebral cortex of SAH. The region marked by rectangular is the region of interest for the following investigation, which is nearby the bleeding region. **c** Intracellular potassium levels in cortical brains of the SAH and Sham mouse model in vivo after 4-AP at the dosage of 1 mg/kg[53,54] immediately and 20 h after SAH and control administration (see "Method", $n = 3$ mice). $p < 0.0001$ (Sham vs. SAH), $p = 0.0005$ (SAH vs. SAH + 4-AP). **d** RBM3 mRNA expression in vivo as measured by qPCR across five groups: Sham, SAH, SAH + 4-AP, SAH + 4-AP + lenti-shRBM3, and SAH + 4-AP + lenti-shNC. Mice were treated with vehicle or 4-AP as indicated, and lentiviral vectors encoding shRBM3 or shNC were administered where applicable (see (**a**) and "Method", $n = 3$ mice). $p = 0.0006$ (Sham vs. SAH), $p = 0.0461$ (SAH vs. SAH + 4-AP), $p < 0.0001$ (SAH + 4-AP+sh-NC vs. SAH + 4-AP+sh-RBM3). **e** Inclusion level of RBM3 exon 3a in the region of interest (**b**) of the SAH and sham mouse model in vivo after 4-AP and control administration. RBM3 Exon 3a expression was quantified with RBM3 exon 3a qPCR primer-1 normalized with mGAPDH ($n = 3$ mice). $p = 0.0003$ (Sham vs. SAH), $p = 0.0066$ (SAH + 4-AP vs. SAH + 4-AP+sh-NC). **f** RBM3 protein expression in the region of interest (**b**) of the mice treated as (**d**), shown as relative RBM3 signal per cell in RBM3 immunostainings ($n = 5$ mice; see Supplementary Fig. 10c). Each dot represents the RBM3 intensity in an individual cell ($n = 13$ cells per slide), with five mice analyzed per treatment group. $p < 0.0001$ in all comparisons. **g** Fraction of apoptotic cells in the region of interest (**b**) of the mice treated as in (**d**) shown by violin plot. The data was quantified from images as shown in the supplementary Fig. 10d ($n = 5$ mice). $p = 0.0007$ (Sham vs. SAH), $p = 0.0275$ (SAH vs. SAH + 4-AP), $p = 0.0047$ (SAH + 4-AP+sh-NC vs. SAH + 4-AP+sh-RBM3). **h** Neuronal count in the region of interest (**b**) of the mice treated as in (**d**). Data was quantified from NeuN signal-positive cells in Supplementary Fig. 10c, d ($n = 10$ mice). $p < 0.0001$ (Sham vs. SAH), $p = 0.0051$(SAH vs. SAH + 4-AP), $p = 0.0013$ (SAH + 4-AP+sh-NC vs. SAH + 4-AP+sh-RBM3). **i** Representative images of HE staining in the region of interest (**b**) of the mice treated as (**d**). Representative healthy neurons are highlighted by black arrows, damaged neurons by red arrows, red blood cells in the capillary by green arrows, healthy glia by arrowheads, damaged glia by yellow arrows, and macrophages by purple arrows ($n = 5$ mice). **j** Spongiosis score in the brain region highlighted in (**b**) of the mice treated as in (**d**), quantified from hematoxylin and eosin (HE) staining (**i**) (refer to "Methods"; $n = 5$ mice). $p < 0.0001$ in all comparison. **k** Healthy neuronal count in the region of interest (**b**) of the mice treated as in (**d**), quantified from hematoxylin and eosin (HE) staining (**i**) ($n = 5$ mice). Healthy neurons are identified by round or oval cell bodies with lightly eosinophilic cytoplasm, centrally located nuclei, and prominent nucleoli. Unhealthy neurons are identified by eosinophilic necrosis, exhibiting cell body shrinkage, intensely stained eosinophilic cytoplasm, and darkly stained pyknotic nuclei. **l** Modified Garcia score assessing neurological function in mice treated as described in (**d**) (see "Methods"[55]). This composite score evaluates spontaneous activity, limb symmetry, forepaw outstretching, climbing, body proprioception, and response to vibrissae stimulation ($n = 10$ mice). $p < 0.0001$ (Sham vs. SAH, SAH + 4-AP vs. SAH + 4-AP+sh-RBM3), $p = 0.0002$ (SAH vs. SAH + 4-AP, SAH + 4-AP+sh-NC vs. SAH + 4-AP+sh-RBM3). **m** Schematic representation of the Rotarod Test (see "Method"). It was generated with BioRender. **n** Latency to fall of the mice treated as in (**d**) in the Rotarod Test (see "Method", $n = 10$ mice). This test assesses motor coordination and balance by measuring the time each mouse remains on a rotating rod before falling. $p < 0.0001$ (Sham vs. SAH), $p = 0.0073$ (SAH vs. SAH + 4-AP), $p = 0.0004$ (SAH + 4-AP+sh-NC vs. SAH + 4-AP+sh-RBM3, SAH + 4-AP vs., SAH + 4-AP+sh-RBM3). The plot of (**c**) is points connecting line with error bars (mean ± SD). **h**, **j** are presented as box-and-whisker plots displaying all individual data points. The box represents the interquartile range (IQR) with the center line indicating the median, and the whiskers extend to the minimum and maximum values. **g** is presented as a violin plot showing all points of biological replicates. Others are the scatter dot plots shown as the mean ± SD. Statistical analysis was conducted using ordinary one-way ANOVA followed by Šídák's multiple comparisons test with a single pooled variance. ns denotes no significance, * denotes $p \leq 0.05$, ** denotes $p \leq 0.01$, *** denotes $p \leq 0.001$ and **** denotes $p \leq 0.0001$. Source data are provided as a Source Data file.

To further illustrate the neuroprotective efficacy of RBM3 against SAH, we augmented RBM3 expression in vivo in mice via lentivirus-mediated RBM3 overexpression prior to SAH (Fig. 7a and Supplementary Fig. 11a). Heightened RBM3 levels were markedly correlated with diminished spongiosis severity (Supplementary Fig. 11b, c), increased healthy neurons (Supplementary Fig. 10d), increased neuronal count with normal morphology (Supplementary Fig. 11e, f). RBM3 upregulation also significantly elevated Garcia scores[58,59] in mice with SAH (Supplementary Fig. 11g) and modestly improved performance in the Rotarod test, as evidenced by increased latency to fall (Supplementary Fig. 11h).

Taken together, our results show that the rG4 near splice sites can serve as mammalian thermosensor to modulate alternative splicing network. Specifically, G-rich sequences around splice sites of RBM3 exon 3a form stable rG4 structures at low temperatures (33 °C and 35 °C) or high potassium concentration that repress exon inclusion, resulting in increased expression of the neuroprotective RBM3 by preventing NMD. This regulation may be exploited in clinical settings of SAH, as we show that potassium-mediated increase in RBM3 acts neuroprotective in vivo.

## Discussion

Temperature has been shown to influence splice site selection by modulating the activity of *trans*-acting factors, whose expression may vary across cell types, which contributes to splicing heterogeneity. Here, we describe a complementary mechanism that relies on the intrinsic structural properties of RNA itself and can therefore function in *cis*. We demonstrate that temperature-controlled formation of rG4 structures near splice sites of cassette exons regulates alternative splicing. Our data reveal that approximately 10–20% of exons repressed upon cold shock harbor putative rG4 motifs (or even up to 35.7% when assessed using G4Hunter) in the vicinity of their splice sites, supporting a widespread role for rG4s as physiological thermosensors in mammalian cells. We propose that temperature-induced rG4 folding modulates splice site accessibility, thereby contributing to temperature-sensitive splicing regulation. While this mechanism can operate through RNA alone, we also acknowledge that RNA-binding proteins (RBPs) play an important modulatory role. RBPs often exhibit structural preferences—some bind to single-stranded regions while others preferentially interact with structured or double-stranded RNA. Thus, temperature-driven changes in RNA conformation may not only alter splice site accessibility directly but also impact the binding affinity and activity of RBPs in these regions. For example, RBPs[40,60–62], such as hnRNPF[40] and hnRNPH[34] may preferentially bind either to stable rG4 structures or to unfolded G-rich single-stranded sequences, thereby contributing to temperature-dependent splicing outcomes. Together, our findings support a model in which the RNA molecule itself serves as a temperature-responsive regulatory element, while the action of RBPs may fine-tune or amplify these RNA-driven effects without being strictly required for the temperature-sensing function.

Based on our in vitro data showing modulation of the rG4 structure in the physiological temperature range, corresponding mutational analysis in cell culture experiments, G4-specific immunostaining, and SHAPE-MaP probing, we classify these sequences as mammalian RNA thermometers. Temperature-controlled behavior of rG4s will be additionally controlled by RBPs in the cellular environment, which may alter the ultimate temperature response of the RNA. However, as the temperature range with dynamic response of the RBM3 rG4 is very similar in vitro (CD and NMR) and in cells (G4-specific immunostaining, SHAPE-MaP, exon skipping and RBM3 expression), we suggest, that at least in this case, the RNA is sufficient to act as temperature sensor and the temperature range is not substantially influenced by cellular RBPs. Since we found that a substantial percentage of CREs contain rG4 motifs, rG4s may serve as widespread

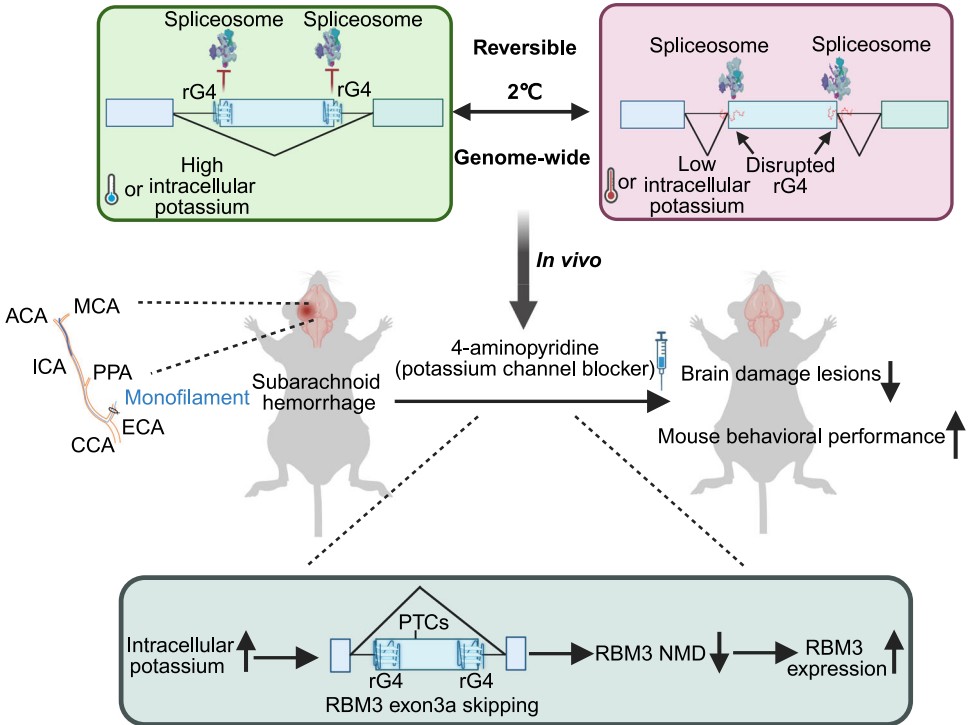

**Fig. 8 | Working model that rG4s function as RNA thermometers to modulate alternative splicing network.** Here is a refined model of RNA G-quadruplexes acting as evolutionarily conserved thermo- and potassium sensors, modulating alternative splicing in mammals. RNA G-quadruplexes (rG4s) function as reversible temperature sensors, impacting alternative splicing dynamics. In low temperatures or high potassium conditions, stabilized rG4s can mask surrounding splice sites, rendering these sites inaccessible, thereby promoting exon skipping. Conversely, at high temperatures or under low potassium conditions, rG4s become destabilized, allowing splice sites to be exposed, and facilitating efficient exon inclusion. RBM3, a well-known cold-induced protein with neuroprotective functions, harbors a poison exon with rG4s around splice sites, that, upon inclusion, triggers NMD (non-sense mediated decay) of the RBM3 mRNA. Under low temperatures or high potassium conditions, rG4s shield the splice sites, leading to poison exon skipping and increased RBM3 expression. Stabilization of these rG4s through increased K+ promotes poison exon skipping, enabling escape from NMD, and ultimately elevating RBM3 expression. Notably, 4-AP, a clinically used pan voltage-gated potassium channel blocker, protects against neuronal damage in a subarachnoid hemorrhage mouse model in an RBM-dependent manner. (ISS intronic splicing silencer, ESE exon splicing enhancer, ACA anterior cerebral artery, MCA middle cerebral artery, PPA pterygopalatine artery, ICA internal carotid artery, ECA external carotid artery, CCA common carotid artery). This figure was generated with BioRender.

physiological thermosensors regulating alternative splicing in mammalian cells, which may be targeted in new therapeutic approaches in diverse conditions (Fig. 8).

The repressive effects of G4 stabilizers on exon inclusion vary across rG4 elements[63], likely due to structural differences or interactions with *trans*-acting factors. G4 ligands (PDS, Phen-DC3, and NMM) exhibit differential binding affinities to rG4s, influenced by backbone orientation, loop structure, and stabilizing elements. Alternatively, this minimal effect of NMM (Supplementary Fig. 3a–h) may reflect a nuanced interplay between the intrinsic thermo-responsive behavior of rG4 structures and the cellular context, including the presence of RNA-binding proteins (RBPs). While rG4s are structurally responsive to temperature changes, their functional impact on splicing likely depends not only on folding per se but also on dynamic competition with RBPs that recognize or modulate G-rich elements. Given that in general NMM binds and stabilizes folded rG4 structures, its limited effect on splicing suggests that RBP-mediated remodeling may compete with or override ligand-induced stabilization. Also, the local structural context or subcellular localization of rG4s may restrict ligand accessibility in vivo. These variations may explain the varying responses of rG4s to stabilizers and highlight the potential for designing selective small molecules that preferentially or specifically stabilize individual rG4s. Such specific rG4-targeted stabilizers could offer a promising therapeutic strategy, enabling specific upregulation of RBM3 while minimizing off-target effects. Such orally available compounds represent a potential neuroprotective approach for hemorrhagic stroke and other neurological disorders.

KCl depolarization influences alternative splicing[64], yet its mechanistic basis remains unclear. Our findings suggest that a subset of K+-responsive exons is regulated via rG4 stabilization, consistent with K+'s role in maintaining stable rG4 structures. Given the critical role of potassium homeostasis in cellular function[65], its disruption in disorders such as kidney disease[65], vascular disease[66], and neurodegenerative conditions[67,68] may drive dysregulation of alternative splicing and gene expression. We show that intracellular K+ elevation stabilizes rG4, modulating splicing and providing neuroprotection in hemorrhagic stroke. The role of 4-AP in related conditions, its RBM3-dependent neuroprotection and its off-target effects warrant further investigation to identify additional molecular targets.

Our findings highlight that 4-AP can mimic, at least in part, the RBM3-inducing effects of therapeutic hypothermia, thereby offering potential as a pharmacological alternative. We noticed that therapeutic hypothermia usually induces 1.8 to 4-fold increase of RBM3 expression[8,10,69,70]. On the other hand, our data show that 24-h 4-AP treatment leads to an approximately 1.5-fold increases in RBM3 mRNA and 3.5-fold increases in protein levels in the hemorrhagic mouse hippocampal HT22 cell model. In the mouse model, 4-AP treatment increases RBM3 mRNA levels by approximately 1.3-fold and protein levels by 1.7-fold. Therefore, the effects of 4-AP at our tested dose induced quite similar RBM3 expression changes compared to therapeutic hypothermia. Furthermore, in this study, 4-aminopyridine (4-AP) was administered only on the first day post-SAH, leaving the potential neuroprotective effects of prolonged treatment unexamined. Additionally, assessing cognitive function in the acute phase

of SAH is particularly challenging, as it requires a recovery period or prolonged observation for accurate evaluation. Future behavioral assessments should also incorporate cognitive function testing to provide a more comprehensive understanding of neurological outcomes of 4-AP. Altogether, our results lay the groundwork for promising new therapeutic avenues for acute brain injury that could benefit from elevated RBM3 expression, and more broadly, for conditions that can be treated with hypothermia, such as neurodegenerative diseases[10].

## Methods

All experimental procedures described in this study were performed in strict accordance with relevant institutional and national guidelines and regulations for research involving animals and cell lines. The animal studies were reviewed and approved by the Institutional Animal Care and Use Committee (IACUC) of Wuhan University (Approval No.: WDRM20240302B). All in vitro experiments, including cell culture, RNA-Seq, molecular cloning, transfection, RNA extraction, RT-PCR/qPCR, SHAPE-MaP probing, biophysical analyses (CD spectroscopy, NMR), immunostaining, protein extraction and western blotting, bioinformatics analyses, cytotoxicity assays (CCK-8 and live/dead staining), and intracellular potassium measurements, were performed following institutional biosafety regulations and standard laboratory practices. No human participants or primary human materials were involved in this study. All key resources used in this study are listed in Supplementary Data 2.

### Cell culture and treatment

HEK293T (CRL-3216™, ATCC), Hela (CCL-2™, ATCC), HEK293 (CL-0001, Procell Life Science & Technology Co., Ltd.), and HT22 (iCell-m020, iCell Bioscience Inc.) cells were cultured on uncoated plastic flasks or plates in DMEM medium ($4.5 \times g$ glucose/L, supplied with GlutaMAX L-glutamine (10566, Gibco) supplemented with 10% fetal bovine serum (Biochrom) and penicillin-streptomycin (1:100 from a stock of 10000 U/mL, 15140122, Gibco) in 5% $CO_2$ at 37 °C. N2a cells (CL-0168, Procell Life Science & Technology Co., Ltd.) were cultured also on plastic flasks or plates in the medium with DMEM/opti-MEM (1:1) supplemented with 10% fetal bovine serum (Biochrom) and penicillin-streptomycin (1:00 from a stock of 10000 U/mL, Gibco, 15140122). At roughly 80% confluency, cells were sub-cultured using a trypsin solution (1:10 dilution of 2.5% stock, 15090046, Gibco) in phosphate-buffered saline. All cell lines were authenticated by the suppliers at the time of purchase (Certificates of Analysis available) and further verified by short tandem repeat (STR) profiling. Routine mycoplasma testing was performed every three months, and all results were negative.

Cells were treated with G4 stabilizer (PDS (HY-15176, MCE), Phen-DC3 (HY-15594A, MCE) (10 μM) at different temperatures for 48 h. Cells were treated with KCl (50 mM) at different temperatures for 24 h (for qPCR) or 48 h (for WB). HT22 cells were treated with different dosages of 4-aminopyridine (4-AP) (HY-B0604, MCE) or amifampridine, and hemin (150 μM) if mentioned together for 24 h. For the RBM3 overexpression assay, HT22 cells were seeded into 12-well plates until the cell confluence reached 20–30%. Then, the medium of each well was replaced with complete culture medium (200 μl) containing 10 μl of virus (at a concentration of $1 \times 10^8$ TU/ml) and 8 μl of HitransG P solution (REVG005, Gekai). After 24 h of culture, the medium was replaced to fresh complete culture medium again. Cell viability and EGFP fluorescence expression were observed under a microscope 72 h after infection. Subsequently, the cells were cultured in complete medium containing 5 μg/ml puromycin for at least 14 days for infection-positive selection, until the proportion of fluorescent cells observed under a microscope reached 100%. The cells were then used for subsequent experiments.

### RNA-Seq

For RNA-Seq, biological duplicates of two independent Hela cell clones (total $n = 4$) were seeded on 6-well dishes and grown for ~48 h at 37 °C to 75% confluence. Cells were then shifted to 32, 37, and 40 °C, and incubated for 24 h. Total RNA from Hela was extracted as described below. Sequencing libraries were prepared by poly(A) selection using the TruSeq mRNA Library Preparation Kit. Sequencing was performed on an Illumina HiSeq 2500 system with V4 sequencing chemistry, generating around 50 million 150-bp paired-end reads per sample. RNA-Seq data are made publically available under GSE262498.

### Cloning

All cloning regarding RBM3 minigenes was done using the ClonExpress II One Step Cloning kit from Vazyme (C112). Briefly, the WT or mutated DNA fragments were generated by PCR, then the PCR fragments and the vector fragments from pcDNA3.1(+), which were digested with XhoI and HindIII enzyme, were incubated together for 30 min. Then, the reaction solution was directly added to DH5apha competent cells on ice for another 30 min, followed by heat shock of around 1 min at 42 °C. Heat-shocked bacteria were immediately put on the ice for 5 min, followed by adding 300 μl of LB medium without antibiotics. These bacteria were then cultured at 37 °C for another 1 h, and then they were plated on the agar plates containing ampicillin. For the MINX minigene cloning, the constructs were ordered as synthetic genes from TWIST in pTWIST-CMV. They contain exon 1 to intron 1 sequence of MINX, followed by the 3' splice site of RBM3 exon 3a (with G4wt or mut). Then, a small part of the MINX exon 2 sequence replaces the E3a sequence, followed by the RBM3 E3a 5' splice site leading into the exon 4 sequence. For the fluorescent minigene, the backbone pCDNA3.1(+) Hygro and the directly full-synthesized sequence ctr and insertion fragment (DINGKE Biotechnology Co., LTD) (see key resources for the sequence) were digested with HindIII and XhoI, then the same procedure with RBM3 minigene cloning was followed.

### Transfection

PEI (HY-K2014, MCE) was used as a carrier to transfect plasmids into cells. Briefly, around 1000 ng plasmid per well (12-well plate) was mixed with 100 μl of opti-MEM, and 3 μl of PEI (1 μg/ml) was added into another tube containing 100 μl of opti-MEM. These two solutions were put at room temperature for ~5 min, and then they were mixed at room temperature for another 20 min. This mix was slowly added to overnight sub-cultured cells. WT or mutant mRBM3 and hRBM3 minigenes were transfected into cells at 37 °C overnight, then followed by different treatments if mentioned for another 24 h at different temperatures.

For siRNA (20 μM) transfection, RNATransMate (E607402, Sangon Biotech) served as the transfection reagent. In brief, approximately 10 μl of RNATransMate and 7 μl of siRNA per well (6-well plate) were diluted with 200 μl of serum-free DMEM medium, respectively. These two solutions were mixed and incubated at room temperature for 10 min to form the RNA/RNATransMate complex. Subsequently, this mixture was added to overnight-subcultured HT22 cells with approximately 60%–70% confluency. The cells were cultured for another 24 h to induce knockdown of the target gene, followed by subsequent chemical treatment.

### RNA extraction, RT-PCR, and qPCR

For the HEK293T and N2a cells, Trizol (BS67.211.0100, Bio&SELL RNA Tri-Liquid) was used for RNA extraction as described in the user manual. Shortly, Trizol was directly added to cells after removal of the medium. Then, 1/5 volume $CH_3Cl$ was added and mixed, followed by $13,000 \times g$ centrifuge for around 30 min. The supernatant was added to the same volume of isopropanol, and then the mixture was centrifuged at $13,000 \times g$ for 30 min to pellet the RNA. The RNA pellet was washed with 70% ethanol (diluted with DEPC-treated water) and was

dried at room temperature for 1 min. The RNA was subjected to DNAse digestion at 37 °C for at least 30 min, followed by PCI extraction and precipitation. Reverse transcription, RT-PCR, and qPCR were performed as described previously[71].

For the experiments of the hemin-induced HT22 cell model and the in vivo mouse model, brain tissues from the affected region (see in vivo mouse experiments) were ground with a grinder on ice and then dissolved in Trizol (R701-01, RNA-easy isolation reagent). For the HT22 cells treated in KCl, 4-AP, and AFP treatment assay, Trizol (Invitrogen) was directly added into the cells after the culture medium was removed. Then, the cell lysates in Trizol were used for RNA extraction, and the procedures were the same as previously mentioned. Then, the cDNAs were generated as follows. For the qPCR of the hemin-induced HT22 cell model and the in vivo mouse model, 1st Strand cDNA Synthesis SuperMix for qPCR kit (11141ES60, YEASEN) was used to produce total cDNAs; for the HT22 cells transfected with minigenes in KCl, 4-AP, and AFP treatment assay, minigene-specific transcripts were generated with the kit from Transgen (AT311, TransScript® One-Step gDNA Removal and cDNA Synthesis SuperMix) according to the manufacturer's instructions with BGH-R primer. Briefly, around 500 ng of total RNA was incubated with the YEASEN or Tansgen Supermix containing reverse transcriptase, cDNA remover, and RT-primer for 30 min at 42 °C, followed by heating for 5 s at 85 °C.

All the primers used are shown in the key resources table. For RT-PCR, the skipping and inclusion bands were amplified by corresponding gene-specific primers, and amplified bands were detected by agarose gels. MINX minigene splicing was analyzed with radioactive RT-PCR using a radioactively labeled forward primer. The RT-PCR data in HEK293T and N2a cells was quantified with ImageQuant TL software. For the RT-PCR data in HT22, Fiji (ImageJ2 V2.16.0/1.54g) was used to quantify the data. For the qPCR experiments in HEK293T and N2a cells, mRNA was normalized using hHPRT and mHPRT, respectively. For the qPCR experiments in HT22 cells and in vivo mouse models, mRNA expression levels were normalized using mGAPDH for qPCR analyses.

## CD spectroscopy analysis

Briefly, the CD (circular dichroism) signal of rG4 was checked using a Jasco J-1500 CD spectrophotometer and a 0.1-cm path length quartz cuvette (Hellma Analytics) was employed using a volume of 200 μl. Samples with 5 μM RNA (final concentration) were prepared in 50 mM KCl or 0.1 mM KCl in 50 mM Tris-HCl pH 7.5. Each of the RNA samples was then thoroughly mixed and denatured by heating at 95 °C for 5 min and cooled to room temperature overnight for renaturation. The RNA samples were scanned from 220 to 310 nm at different temperatures and spectra were acquired every 1 nm. All spectra reported were an average of 3 scans with a response time of 0.5 s/nm. They were then normalized to molar residue ellipticity. For measurements at different temperatures, each sample was equilibrated at the designated temperature for approximately 30 min before measurement.

## NMR

All 1D $^1$H spectra were recorded on a Bruker Avance III spectrometer equipped with a $^1$H/$^{13}$C/$^{15}$N TCI cryoprobe. The sample temperature was adjusted with a MeOD-d4 sample according to Findeisen et al.[72]. The RNA sample (Phosphate-GGGGGUGGAGGGCAGCGGCAGA) was synthesized by Microsynth on a 0.2 μmol scale giving a yield of 23.7 nmol. The RNA was dissolved in 500 μl of a 95% $H_2O$/5% $D_2O$ (100 atom%D, ARMAR Chemicals, 1070-1 × 10ML) solution containing 2 mM KCl or 0 mM KCl, yielding a concentration of 47 μM. A standard 5 mm NMR tube (TA, ARMAR Chemicals) was used. 1D $^1$H spectra were recorded using a gradient 1−1 echo sequence[73] with the application of two pulse field gradients of 1 ms and 15.9 G/cm similar to the gradient-enhanced 11 echo HMQC[74]. Pulse train delays of 45 μs and 90 μs were used for the first and second pulse pair, respectively. Spectra were recorded with 8192 complex points, a spectral width of 22.0 ppm,

typically 512 transients, and a relaxation delay of 1.5 s. The irradiation frequency was adjusted to the remaining water signal. Spectra were processed using a quadratic sine shifted by 90°. Chemical shifts were referenced to DSS (2,2-Dimethyl-2-silapentane-5-sulfonate sodium salt) using an external commercial standard sample for testing water suppression containing 2 mM sucrose and 0.5 mM DSS (Bruker Biospin), measured at exactly the same temperatures.

## Immunostaining

For G4-specific immunostaining, cultured cells were seeded on glass coverslips and incubated at different temperatures for 48 h. After treatment, cells were fixed with 4% paraformaldehyde (PFA) for 15 min and permeabilized with 0.5% Triton ×-100 for 10 min. Blocking was performed using 5% horse serum in PBS for 1 h, followed by treatment with DNase (EN401, Vazyme) or Benzonase (70664, Merk) for 30 min. Cells were then incubated with a G4-specific antibody (1:500, Ab00174-30.126, Anti-DNA/RNA G-quadruplex [BG4], Absolute Antibody)[42–44] for 1 h, followed by an anti-FLAG secondary antibody (1:1000, 637317, Alexa Fluor® 488 anti-DYKDDDDK Tag Antibody, BioLegend) and Hoechst incubation for another 1 h. Finally, stained cells were mounted on glass slides using mounting medium (03989, Sigma).

For other immunostainings, After the cultured cells were cleaned with PBS, HT22 cells were fixed with PFA for 10 min and incubated with 0.25% Triton-× 100 for 30 min. The sections were dewaxed, rehydrated, boiled in pH 6.0 citrate antigen retrieval solution (G1206, Servicebio) and incubated with 0.25% Triton-× 100 for 30 min. After blocking with 5% bovine serum albumin (BSA) for 1 h, HT22 cells and the sections were directly incubated with the primary antibodies (mouse anti-NeuN (66836-1Ig, Proteintech) and rabbit anti-RBM3 (14363-1-AP, Proteintech)) at 4 °C overnight. After washing with PBS, the cells were incubated with 5% BSA containing the Alexa 488-or 594-conjugated secondary antibodies. After washing again, the cells and sections were immersed in 4′, 6 diamidino-2-phenylindole (DAPI) for 30 min. Immunofluorescence images were acquired with a fluorescence microscope. To detect neuronal death, double immunostaining of NeuN (66836-1-Ig, Proteintech) and TUNEL was conducted by TUNEL Assay Kit (C1089, Beyotime) in coronal sections at 24 h after SAH according to the manufacturer's instructions.

## SHAPE-MaP probing

HEK293 cells were treated with a G4 stabilizer (PDS (HY-15176, MCE) or Phen-DC3 (HY-15594A, MCE)) or DMSO for 24 h or incubated at different temperatures for 48 h. Following treatment, cells were incubated with 50 mM NAI (N288069, Aladdin) for 15 min, followed by 0.5 M DTT (HY-15917, MCE) incubation for an additional 10 min. Chromatin-associated RNA was then extracted[75]. The extracted RNA was reverse-transcribed into cDNA in an $Mn^{2+}$-containing buffer using Superscript II reverse transcriptase (18064071, Invitrogen)[45] and a gene-specific primer with random sequence that binds around 50 bp downstream of the putative rG4. The resulting cDNA was amplified using a high-fidelity DNA polymerase (p505, Vazyme) with PCR primer pairs specifically designed to amplify the region encompassing around 50 bp upstream and downstream of the putative rG4, resolved by gel electrophoresis, purified, and subjected to Sanger sequencing. Indel percentages were determined using TIDE analysis[49] within the 50 bp upstream and downstream regions of the putative rG4.

## Protein extraction and western blot

For the HEK293T and N2a cells, whole-cell extracts (WCEs) were prepared with lysis buffer (20 mM Tris (pH 8.0), 2% NP-40 (v/v), 0,01% sodium deoxycholate (w/v), 4 mM EDTA, and 200 mM NaCl) supplemented with protease inhibitor mix (Aprotinin, Leupeptin, Vanadat, and PMSF). For the HT22 cells and brain samples from mice, WCEs were prepared with IP lysis buffer (G2038, Servicebio) containing cocktail protease inhibitor (G2006, Servicebio) with sonication.

Protein concentrations were determined using BCA assay (23225, Pierce™ BCA Protein Assay Kit, Thermofisher) according to the manufacturer's instructions. SDS-PAGE and Western blotting followed standard procedures. Western blots were quantified using the ImageQuant TL software (HEK293T and N2a cells normalized with hnRNPL) and Fiji (ImageJ2 V2.16.0/1.54 × g) (HT22 cells and mice normalized with GAPDH). The following antibodies were used for Western blotting: hnRNPL (4D11, Santa Cruz), RBM3 (14363-1-AP, Proteintech), GAPDH (A19056, Abclonal), and NeuN (66836-1-Ig, Proteintech).

## Bioinformatics analysis

To reveal temperature-dependent changes in exon inclusion, 150 nt paired-end samples from HEK293T and HeLa were aligned to the human hg38 genome, using STAR (V2.6.1a). Exon inclusion levels were then calculated using rMATS (V3.1.0) and further filtered using standard Python code (Jupyter Notebooks with Python version 3).

Cassette exons (Percent Spliced-In (PSI) > 0.05 and PSI < 0.95, and exon length > 20 bp) were classified into three groups, including cold-repressed exons (CRE), cold-induced exons (CIE), and not temperature-sensitive (NS), based on the changes of PSI (delta PSI) values at different temperatures. In detail, for HEK293T cells, CRE and CIE were defined as those exons with | delta PSI| ≥ 0.1 and BH-adjusted $p$ value < 0.05 in the comparison of cells at 35 °C vs. 39 °C. For Hela, CRE and CIE were defined similarly in two comparisons, 32 °C vs. 37 °C, and 37 °C vs. 40 °C, respectively. However, the final NT exons were defined by the intersection of NT in these two comparisons (Fig. 1d). The sequence flanking each cassette exon, and the upstream and downstream exons (Fig. 1b) were extracted using bedtools (V2.3)[76]. We further predicted G4 in these sequences by searching motifs from a previous study[27], and using G4Hunter[28]. The G4 motifs were searched in 50 bp sequences flanking splice sites. For G4Hunter analysis, we scanned 200 bp sequences flanking splice sites with a 25 bp window at each base to get a nucleotide-resolution score. All exons were used to examine the general correlation between G4 score and delta PSI. Hela rG4 sequencing (rG4-seq) data were downloaded from the Gene Expression Omnibus (GEO) with accession number GSE77282. The identified rG4 regions with a stop signal in cells cultured in the presence of K$^+$ or PDS (K$^+$ and PDS) were converted from hg19 to hg38 coordinates with UCSC LiftOver and used for the downstream analysis. We calculated the number of exons overlapped with rG4 peaks at each position from −200 to 200 nt for cassette exons and smoothed it with a 30 nt window. To compare the rG4-seq peak frequency across the three categories of exons (CRE, CIE, and NT), the Reverse Transcriptase Stalling (RTS) values of the overlapped peaks were accumulated separately for each category. To obtain consistently temperature-regulated exons in HEK293T and Hela, we relaxed the thresholds to define CRE and CIE by requiring BH-adjusted $p$ value < 0.05 and delta PSI > 0 (CIE) or < 0 (CRE).

For reproducing results/figures in the RNA thermometer study, the original R (V4.2.1) codes are available in GitHub (https://github.com/christear/G4splicing). For the splice site strength prediction, we used MaxEntScan (http://hollywood.mit.edu/burgelab/maxent/Xmaxentscan_scoreseq.html) to get the 5′-splice site strength. For sequence alignment, we employed DNASTAR (V17.3.0.57) to generate the alignment results. The schematic figures in this paper were made by BioRender (https://www.biorender.com/).

## CCK-8 assay and dead/live staining

HT22 cell viability was measured using the cell counting kit-8 (CCK-8) according to the manufacturer's instructions (HY-K0301, MCE). Briefly, 2000 cells were seeded in 96-well plates and allowed to grow overnight before treatment with the indicated concentrations of 4-AP for 24 h. After treatment, the cell culture medium was replaced with 100 µL fresh medium containing 10 µL CCK-8 solution and incubated for 1–3 h in a humidified incubator. The absorbance at 450 nm was measured with a microplate reader. For the dead/live cell assay, live cells were stained with LiveDye (a cell-permeable green fluorescent dye), and dead cells were stained with NucleiDye (a cell non-permeable red fluorescent dye). In the dead/live staining (KTA1001, Abbkine), after treatment with 4-AP for 24 h, HT22 cells were washed with PBS and then incubated in buffer solution with LiveDye and NucleiDye for 30 min darkness. Cells were washed again with PBS, and the fluorescence was observed under a fluorescence microscope.

## In vivo mouse experiments

The Animal Center of Wuhan University provided 20−25 × g adult male C57BL6/J mice. They were housed in standard conditions of 22 °C and 50−60% relative humidity with a 12 h light/dark cycle and free access to food and water. The SAH mouse model was performed by the intravascular perforation method. Mice were anesthetized with isoflurane. The left internal carotid artery (ICA) was exposed in the median neck incision. A silicone-coated monofilament was inserted from the left external carotid artery (ECA) along the ICA to the bifurcation of the anterior cerebral artery (ACA) and middle cerebral artery (MCA). When resistance was encountered, continue to push forward by 1 mm to puncture the blood vessel. Then monofilament was removed and the ECA was ligated. The sham group underwent the same surgical procedure without an endovascular puncture. Mice were intraperitoneally injected with 4-aminopyridine (4-AP) (HY-B0604, MCE) immediately at the dosage of 1 mg/kg[53,54] after SAH and 20 h after SAH. Mice were stereotaxically microinjected with lentiviral particles (GeneChem Co., Ltd.) containing lentivirus overexpressing RBM3 (OE-RBM3) (GeneChem Co., Ltd., NM_001166410) or a negative control (OE-NC) (GeneChem Co., Ltd.) and lentivirus sh-RBM3 (GTTGATCATGCAGGAAAGTctcgagAGACTTTCCTGCATGATCAAC) or sh-NC (GTCTCCACGCGCAGTACATTTctcgagAAATGTACTGCGCGTGGAGAC) at a dosage of 2.0 µl (5 × 108 TU/mL) into the left cerebral cortex 7 days before SAH onset. The stereotaxic coordinates are as follows: point 1, anteroposterior 0.3 mm from the bregma, mediolateral 3 mm from the midline, and dorsoventral 4 mm from the skull; point 2, anteroposterior 0.8 mm from the bregma, mediolateral 3 mm from the midline, and dorsoventral 4 mm from the skull. The injection speed was 0.4 µL/min, and the needle was left in place for 5 min after the injection. For the Hematoxylin and eosin (HE) and Nissl staining, the mice were deeply anesthetized 24 h after SAH and transcardially perfused with cold PBS followed by 10 ml of 10% paraformaldehyde. Whole brains were collected, fixed in 10% paraformaldehyde for 24 h, and then sectioned into 10-µm-thick coronal slices. For the samples of qPCR and WB, the mice were sacrificed by neck dissection immediately after transcardiac perfusion with cold PBS, and the affected cerebral cortex on the ipsilateral side of the bleeding site was removed and placed in a tube and immediately stored at −80 °C for WB or were frozen in liquid nitrogen for qPCR.

## Mouse behavior test

Neurological function was detected using the modified Garcia score and the rotarod test at 24 h after SAH in mice. The modified Garcia score included six standards: spontaneous movement, limb movement, forelimb extension, climbing, tapping one limb, and response to facial irritation, each scoring 0–3. The total score ranges from 3 to 18 (see the table below). In the rotarod test, briefly, mice were placed on a rotating drum with a speed accelerating from 5 to 30 rpm within 2 min. One day in advance, mice were trained on a rotating rod until a stable performance level was reached. After 24 h of SAH, the mice were tested with an initial speed of 5 rpm/min accelerated to a maximum of 30 rpm/min. The duration of animals on the accelerating rotarod was recorded. High neurological test scores indicated better neurological

function. All neurobehavioral tests were performed blinded. The modified Garcia Score table, including expanded descriptions for each component[55], is provided in Supplementary Table 1.

## Hematoxylin and eosin (HE) and Nissl staining

Brain sections were embedded in paraffin and cut into 10 µm-sections. The sections were dewaxed, rehydrated, and stained with haematoxylin and eosin (C0105, Beyotime) or Nissl stain solution (G1036, Servicebio) according to the instructions. Scoring system for spongiosis evaluation: 0: absent spongiosis (no intercellular edema or vesicles/bullae); 1: almost no spongiosis (almost no intercellular edema or vesicles/bullae); 2: mild spongiosis (minimal intercellular edema, few small vesicles/bullae); 3: moderate spongiosis (moderate intercellular edema, multiple vesicles/bullae); 4: Severe spongiosis (marked intercellular edema, numerous large vesicles/bullae). The number of neurons with normal morphology and intensity of Nissl substance was measured by ImageJ.

## Intracellular $K^+$ measurement

After treatment with 4-AP, the culture medium was removed, and the cells were washed with PBS and lysed in deionized water with sonication. 24 h after SAH, the cerebral cortex was taken and fully ground with deionized water. The protein concentration of the homogenates was quantified by a BCA protein assay kit. The $K^+$ concentration was measured by a potassium ($K^+$) turbidimetric assay kit (E-BC-K279-M, Elabscience) according to the manufacturer's protocol. In brief, 80 µL homogenate was mixed with an equal volume of commercial protein precipitant, and after being centrifuged at $1100 \times g$ for 10 min, 50 µL supernatant was added to a 96-well plate, and then 200 µL chromogenic agent was added to the wells and mixed fully with the supernatant. After being incubated at room temperature for 5 min, the optical density at 450 nm was measured with a microplate reader. The $K^+$ concentration was normalized with the protein concentration in the homogenate lysates.

## Statistical analysis

All image quantifications were performed using ImageJ unless otherwise specified. Figure 7h, j and Supplementary Figs. 10f and 11c, f are presented as box and whisker plots with min to max showing all points of biological replicates. Figures 6a and 7c are points connecting line with error bars (mean ± SD). Figure 7g is presented as a violin plot showing all points of biological replicates. The others are presented as scatter dot plots (line at mean ± SD). All individual data points are shown. Except for the bioinformatics analyses in Fig. 1, statistical significance was assessed using GraphPad Prism (V10.4.1 (532)), employing two-tailed Student's $t$-tests or one-way ANOVA followed by Dunnett's or Šídák's multiple comparisons tests with a single pooled variance. Statistical significance was accepted at $p < 0.05$. In the figure legends, "ns" denotes $p \geq 0.05$, * denotes $p < 0.05$, ** denotes $p < 0.01$, ***$p < 0.001$, and **** denotes $p < 0.0001$. Effect sizes were reported as fold enrichment values relative to the control group, representing the magnitude of changes between conditions. Unless otherwise specified, 95% confidence intervals (CI) were calculated based on the underlying statistical test (e.g., two-sided Student's $t$-test, ANOVA, two-sided Wilcoxon test, likelihood-ratio test, or hypergeometric test) using default parameters in the respective statistical software.

## Reporting summary

Further information on research design is available in the Nature Portfolio Reporting Summary linked to this article.

## Data availability

The data supporting the findings of this study are available from the corresponding authors upon request. Source data for the figures and Supplementary Figs. are provided as a Source Data file. Source data are provided with this paper.

## Code availability

Original codes for the rG4 analysis are available in GitHub (https://github.com/christear/G4splicing).

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

## Acknowledgements

We appreciate the discussion and some help from Mary Kennedy, Luiza Zuvanov, and other lab members in Prof. Florian Heyd group at Freie Universität Berlin. The circular dichroism (CD) spectroscopy analysis was performed in the lab of Prof. Dr. Beate Koksch at Freie Universität Berlin with help from Zeinab Mahfouz. The authors thank Peter Schmieder and Monika Berbaum (FMP Berlin) for providing NMR pulse sequences and Jana Sticht (FU Berlin) for NMR data set templates. We also appreciate Prof. Kwok Chun Kit (City U Hong Kong) for his valuable comments in SHAPE-MaP probing. The authors also appreciate the assistance of Junbo Yi (Shenzhen University) with confocal imaging. The study was supported by National Natural Science Foundation of China (No. 81971870 and No. 82172173 to M.L.), the Deutsche Forschungsgemeinschaft (grant HE 5398/4-2 to F.H.), Macau FDCT (Funding Scheme for Scientific Research and Innovation [0167/2023/RIA3] to A.C.) and Shenzhen Clinical Research Center for hematologic disease (LCYSSQ20220823091401002 to L.Y.).

## Author contributions

F.H., M.Z., and M.L. led and supervised the study. M.Z. conceptualized the study with B.Z., F.H., C.L., and M.P. M.Z. and C.L. performed most of the wet lab experiments and data analysis. B.Z. did the bioinformatics analysis of rG4. M.P. performed the RNA-Seq analysis and radioactive PCR. M.A. did NMR experiments. M.S. supervised the NMR experiments. D.L. performed some of in vivo mouse experiments. A.E. and S.M. prepared the samples for RNA-Seq. M.L., W.L., S.C., L.W., L.Z., Q.L., Q.H., R.G., and L.Y. provided advice and insights for neuronal damage in cells and in vivo mouse experiments to mimic clinic scenario. L.Z. helped to analyze HE staining. T.H. contributed valuable insights and discussions to this project. M.Z., B.Z., and F.H. wrote the manuscript with the help from C.L., M.P., M.S., Y.H., M.L., X.Q., A.C., T.W., and X.G.

## Funding

## Competing interests

A patent application (China patent No. 2024105567784) has been filed in relation to this research. There are no other competing interests.

## Additional information

¹Institut für Chemie und Biochemie, RNA Biochemie, Freie Universität Berlin, Berlin, Germany. ²Guangdong Key Laboratory for Biomedical Measurements and Ultrasound Imaging, National-Regional Key Technology Engineering Laboratory for Medical Ultrasound, School of Biomedical Engineering, Shenzhen University Medical School, Shenzhen University, Shenzhen, China. ³Department of Hematology and Oncology, Shenzhen University General Hospital, International Cancer Center, Hematology Institution, Shenzhen University Medical School, Shenzhen University, Shenzhen, China. ⁴Computer Science Program, Computer, Electrical and Mathematical Sciences and Engineering Division, King Abdullah University of Science and Technology (KAUST), Thuwal, Kingdom of Saudi Arabia. ⁵Center of Excellence for Smart Health (KCSH), King Abdullah University of Science and Technology (KAUST), Thuwal, Kingdom of Saudi Arabia. ⁶Center of Excellence on Generative AI, King Abdullah University of Science and Technology (KAUST), Thuwal, Kingdom of Saudi Arabia. ⁷Department of Neurosurgery, Renmin Hospital of Wuhan University, Wuhan, China. ⁸Institut für Chemie und Biochemie, Freie Universität Berlin, Berlin, Germany. ⁹Department of Hematology, Union Hospital, Tongji Medical College, Huazhong University of Science and Technology, Wuhan, China. ¹⁰Department of Pathogen Biology, School of Basic Medicine, Tongji Medical College, Huazhong University of Science and Technology, Wuhan, China. ¹¹State Key Laboratory for Diagnosis and Treatment of Severe Zoonotic Infectious Diseases, Huazhong University of Science and Technology, Wuhan, China. ¹²Department of Pathophysiology, School of Basic Medicine, Chongqing Medical University, Chongqing, China. ¹³Department of Neurology, Union Hospital, Tongji Medical College,

Huazhong University of Science and Technology, Wuhan, China. [14]Department of Cardiovascular Surgery, Union Hospital, Tongji Medical College, Huazhong University of Science and Technology, Wuhan, China. [15]Department of Pediatrics, Union Hospital, Tongji Medical College, Huazhong University of Science and Technology, Wuhan, China. [16]Department of Pathology, Union Hospital, Tongji Medical College, Huazhong University of Science and Technology, Wuhan, China. [17]Department of Rheumatology and Immunology, Union Hospital, Tongji Medical College, Huazhong University of Science and Technology, Wuhan, China. [18]Hubei Engineering Research Center for Application of Extracelluar Vesicles, Hubei University of Science and Technology, Xianning, China. [19]Department of Rehabilitation Medicine, Union Hospital, Tongji Medical College, Huazhong University of Science and Technology, Wuhan, China. [20]Geriatric Medicine Center, Department of Geriatric Medicine, Zhejiang Provincial People's Hospital (Affiliated People's Hospital), Hangzhou Medical College, Hangzhou, China. [21]School of Materials Science and Engineering, Zhejiang Sci-Tech University, Hangzhou, China. [22]Department of Biomedical Sciences, Faculty of Health Sciences, University of Macau, Macao SAR Taipa, China. [23]Ministry of Education Frontiers Science Center for Precision Oncology, University of Macau, Taipa, Macao SAR, China. [24]Department of Medical Neuroscience, School of Medicine, Southern University of Science and Technology, Shenzhen, China. [25]These authors contributed equally: Min Zhang, Bin Zhang, Chengli Liu. ✉e-mail: minzhang900204@gmail.com; mingcli@whu.edu.cn; florian.heyd@fu-berlin.de

