## [Transparent Peer review file · Nature Communications]

Stabilizing a mammalian RNA thermometer confers neuroprotection in subarachnoid hemorrhage

Corresponding Author: Dr Min Zhang

Version 0:

Reviewer comments:

Reviewer #1

(Remarks to the Author)

This is an interesting manuscript where the authors investigate the role of intronic G4s as potential thermometers to regulate alternative splicing in mammal cells in response to temperature changed. G4s have been linked to splicing in the past and they are known to affect alternative splicing, the novelty here is that the authors present evidence suggesting that the fluctuation between a folded or unfolded state of a G4 regulated by changes in temperature might be leveraged to regulate differential splicing as a response of temperature shock. The authors use mutational analysis, PDS treatment and K⁺ alteration to prove this point. They provide some interesting insights presented in this manuscript, but I am afraid that the data generated are all mainly indirect and not sufficient, in the current form, to prove a direct cause/effect between G4-folding/unfolding with temperature in cells to splicing alteration. The main limitation currently is that their biophysical data cannot really be reconciled with the cellular evidence and there is a lack of evidence demonstrating changes in the G4-landscape in living cells upon temperature changes. If the authors can address these main issues I would be supportive of publication in Nature Communication. Below I articulate specific points to be addressed:

1) The folding/unfolding demonstrated by CD in vitro is very modest and, moreover, performed at K⁺ concentrations that are not physiologically relevant. I would strongly doubt that small changes in the CD like the one presented are representative of a significant fluctuation of folded/unfolded G4 at 50mM KCl in such a small and low temperature range, given that especially RNA G4s under physiological K-concentrations are highly stable with melting temperatures above 70 C. The authors use NMR to further prove this point, but it looks like no K at all was used for NMR experiments from the experimental section and therefore this cannot be considered representative of how the actual fluctuation of the structure under physiological conditions with more than 100mM K concentration would look like. An ideal experiment to address this would be to perform in vitro splicing on a construct containing or not the G4 at different temperatures and demonstrate that under physiological conditions the splicing is only affected when the G4 is present but not in a control template.

2) Similarly, in cells the authors provide compelling evidence of splicing being altered as a function of PDS treatment, K⁺-channel blockage at different temperatures, speculating that the G4 stabilisation provided by either PDS treatment or K⁺ justifies the different results obtained a different temperature in the presence or in the absence of these perturbative agents. However, PDS data are really context dependent, and the authors speculate that this is due to the different interactome that PDS disrupt at different sites, which can be a factor. However, PDS is also a DNA damaging agent and is hard to disentangle the effects driven by G4 targeting to response to DNA damage. Moreover, the authors have no evidence like immuno fluorescence for example to demonstrate changes in G4 prevalence at different temperatures and under different conditions. Hence, it hard to reconcile with confidence that changes in temperature affect splicing in a G4-dependent fashion.

To address this it would be essential to have splicing data in cells with more than one G4 ligand (i.e. commercially available PhenDC3 and NMM), to rule out any possible ligand-limited effect (i.e. DNA damage for PDS). Evidence of differential G4 formation at identified sites by immunoprecipitation or global changes by immuno fluorescence would also be ideal to demonstrate a G4-dependent splicing effect.

Providing that the authors can address experimentally these issues I would be supportive of publication in Nat Commun.

(Remarks on code availability)

Reviewer #2

(Remarks to the Author)

Zhang et al. discovered a significant enrichment of rG4 across splice sites of cold-repressed exons. In both human cells and plants, rG4s were previously found to be enhanced in response to cold stress (Kharel et al., 2023 ; Yang et al., 2023). Thus, it is critical in determining the folding status of rG4 in vivo. The authors need to determine the rG4 folding in different temperatures and with/without potassium channel blocker using either DMS or SHAPE-based chemical probing methods (Kharel et al.,2023).

(Remarks on code availability)

NA

Reviewer #3

(Remarks to the Author)

In this study, Zhang and colleagues investigate the significance of RNA G-quadruplex (rG4s) enrichment around alternatively spliced temperature-sensitive cassette exons. They posit that the cold-induced stabilization of rG4s surrounding splice sites prevents the inclusion of cassette exons into the mature mRNA transcript. Using the neuroprotective RBM3 gene as a model, they suggest that the temperature-dependent stabilization of rG4s around a poison exon 3a prevents the inclusion of said exon in the mature transcript, preventing it from undergoing non-sense mediated decay (NMD). They also determine that treatment of cells with potassium, an endogenous rG4 stabilizer, can mimic the effects of low temperature on alternative splicing and thus increase RBM3 expression at normothermia. Using a cellular and in vivo model of subarachnoid hemorrhage, they test the hypothesis that 4-AP, an FDA-approved potassium channel blocker, can mimic the effects of therapeutic hypothermia by increasing intracellular potassium concentrations and thus expression of RBM3, affecting cellular outcomes in the brain and animal behavior.

Although the experiments in this study are generally well-conceived, this study needs significant improvement to definitively reach the conclusions proposed and to convey the results adequately:

Major Points:

- The connection between therapeutic hypothermia, RBM3, subarachnoid hemorrhage, potassium, and rG4s is not well-explained in the introduction section. A few of these concepts are explained thoroughly (e.g., therapeutic hypothermia & subarachnoid hemorrhage) but seem disparate in their presentation and the reader is left to infer the connection. The focus of the introduction should be refined substantially. Certain details are excessive, such as the current state of treatment for subarachnoid hemorrhage, and do not contribute significantly to understanding the study.
- The premises of experiments and descriptions of results need to be explained or clarified prior to results being presented. For example, the relevance and role of potassium is unclear until much later in the results. Similarly, the concept of glutamate stimulation mimicking brain injury is suddenly introduced into the results text with no supporting context, background, or citations. There are several other such instances throughout the manuscript that need to be carefully reviewed by the authors, restructured, and presented in a logical manner.
- Figures need to be reorganized significantly. Certain panels do not convey useful information for understanding the results (e.g. Figs. 1a, 1b, S1a, S1b, S1f, and others). Figure legends in many cases are not adequate and should describe all features of the corresponding figure (i.e., abbreviations, axes, annotations of gels and datapoints on panels, etc.); please carefully review figure legends to ensure the descriptions are accurate and correspond to the proper panel, for example in Fig. 2 panels H and I seem to be flipped. In addition, Fig. 7 which presents the working model of this study would be better presented as an illustration or graphic in the discussion section, not the results section.
- In Fig. 1f and Fig. 1g, the authors show there is a significantly higher G-content and greater presence of G4 motifs at specific regions surrounding cold-repressed exons (CREs). However, there is an apparent difference in G-content and G4 motifs between cold-induced exons (CIEs) and non-temperature sensitive exons (NTs), but these differences have not been statistically tested and/or presented. Furthermore, Fig. 1h, Fig. 1i, and Fig. 1j also support the depletion of G-content and G4 motifs surrounding CIEs. The manuscript focuses on CREs but does not adequately explain the relevance of this depletion surrounding CIEs or mention its relevance in the proposed mechanism.
- The premise of 4-AP treatment seems to be that it can be used to mimic the effects of therapeutic hypothermia on neuroprotective RBM3 expression. The study should establish that RBM3 is fact induced in response to therapeutic hypothermia in the brain and has similar effects on cellular/behavioral outcomes following subarachnoid hemorrhage. Based on the current results, it is unclear whether the effects of 4-AP at the tested dose induces changes in RBM3 expression that are similar or more/less extreme than would be induced by therapeutic hypothermia.

- The sample size for several assays needs to be increased. The current sample size of n=3-4 used in many of the in vivo readouts is not sufficient, especially given well-known animal to animal variations combined with a treatment paradigm that necessitates a much larger sample size for appropriate analysis. This is particularly important when quantifications are made based on histochemical assays and/or when standard deviations are high (such as in Fig. 4K, Fig. 5b, Fig. 5i, etc).
- The formation of rG4 structures in the mouse brain is not completely convincing. The establishment of rG4 folding seems to be based largely on in-silico analysis. While the authors do perform CD/NMR, the folding of the tested fragments outside of the cellular environment is not direct evidence of rG4 folding in the cellular environment, let alone within the brain. Specifically, the use of short fragments in CD/NMR precludes the consideration of the extended sequence of the full-length mRNA, or cellular features, that could affect folding and rG4 formation in vivo. Also, the authors attempt to further prove rG4 formation using rG4-seq data from another study that was conducted in HeLa cells. The reader is led to believe that this extends to the formation of rG4s in the brain, however they do not present a direct validation of this in the current study. The entire premise of the mechanism is based on rG4 formation, so biochemical assays to confirm this folding in the brain are critical. Based on the current data, rG4 formation in the brain cannot be assumed. Further experiments, including rG4-seq or SHALiPE may be used to characterize these structures and their presence in the native environment, which would lend credence to their existence and relevance in the brain and in this particular model.
- Given the therapeutic relevance of 4-AP, dose-dependent effects on RBM3 expression and phenotypic/cellular outcomes are critical. A justification for the current dose is not described, neither is a dose-dependent characterization of the aforementioned readouts presented in this study.

Minor Points:

- The results section begins with a summary of a previous study. It is unclear which findings are specific to the current study. If the results in the first paragraph are taken directly from a previous study, they should be mentioned only saliently and as needed in the results section or summarized in the introduction. Previous results should not be associated with figures unless previous data is being reanalyzed in a novel way.
- Figures and panels should be referred to in the order in which they appear throughout the manuscript.
- The lack of clarity about definitions/abbreviations, naming conventions, and pertinent methodological details in the results section severely hinders the readability of the results. For example, four regions are delineated with undefined abbreviations in Fig. 1C but not explained in the results or the legend. There are several other examples of this, including the naming convention for the mutants/minigenes.
- Given that several cell types were used in this study, they should be consistently mentioned for clarity (e.g., in the hemin-induced model results, it is unclear what cell type is being used until well into the corresponding text).
- The authors make several claims in the results that are based on overinterpretation or speculation. For example, they establish the conservation of G-enrichment in cassette exons broadly in mammals and suggest that this implies a role in splicing events during hibernation (presumably to establish a connection with temperature regulation?), although it is widespread in mammals that do not hibernate, thus one is confused about the link between G-enrichment and hibernation as it could serve a function that is independent of hibernation. Separately, they suggest that increased intensity of Nissl staining implies increased protein synthesis, which is not substantiated by other confirmatory assays. These points may be expanded upon in the discussion but should be removed from the results.
- Statements that make claims not supported by data should be accompanied by citations (multiple in some cases). Several claims throughout the manuscript are not qualified with citations.
- The mechanism proposed is posited as an 'alternative mechanism' to CLK-dependent alternative splicing, however, their results show that G-enrichment around exons occurs in both CLK-dependent and CLK-independent spliced genes. This is confusing and I am not convinced that this is necessarily an alternative mechanism.

(Remarks on code availability)

Version 1:

Reviewer comments:

Reviewer #1

(Remarks to the Author)

The authors have now addressed most of the technical concerns I have raised and I am now supportive of publication in Nat Commun.

The only minor detail to fix prior to publication is to specify the antibody used for G4 IF (BG4), this is mentioned in the

methods but not in the figure caption (please add this for clarity). There are many G4-specific antibodies and the authors need to be more explicit on what they have used. Also the seminal papers where the antibody has been used and developed should be cited.

(Remarks on code availability)

Reviewer #2

(Remarks to the Author)

The authors did not correctly perform the experiments to address my comment. The SHAPE-based method captures the preferential modification of NAI on the last G of every G tract in response to potassium or rG4 ligand. Either SHAPE-induced reverse transcription stalling or mutation occurs at the last G of every G tract. This modification pattern is very specific. The indel surrounding the G-rich motif is not specific to rG4. It is more likely due to the single-stranded nucleotide modified by NAI. Alternatively, the authors could also use the DMS-based method described in Kharel et al., 2023, which is based on the DMS modification on the N7G position, where rG4 formation prevents DMS modification of the N7G position.

(Remarks on code availability)

N/A

Reviewer #3

(Remarks to the Author)

In their revised version of "Stabilizing a mammalian RNA thermometer confers neuroprotection in subarachnoid hemorrhage" Zhang et al. provide compelling mechanistic evidence for the role of RNA G-quadruplexes (rG4s) functioning as temperature-sensitive regulatory elements that modulate alternative splicing. Their revision version incorporates several new assays, including minigene reporter systems, rG4 immunostaining, and SHAPE-based techniques to assess the stability of rG4 folding in vivo, addressing the primary scientific critique of all three reviewers in their prior version. Together, these additions strengthen the claim that the structural stability of rG4s near the splice sites of the poison exon 3a in RBM3 is modulated by changes in temperature and intracellular potassium levels, thus governing RBM3's capacity for expression and eventual translation. Given RBM3's established protective role in the brain, this study identifies a potential therapeutic target for modulating outcomes in neurological injury.

The experimental evidence presented is strong, but we hope the authors consider refining the discussion to acknowledge key caveats of their study/techniques, and do a thorough evaluation of certain details in their manuscript to ensure it is ready for publication.

Minor Points:

1. Of note, in the revised manuscript, the authors conclude that the RBM3 rG4 acts as an autonomous thermosensor and that "the temperature range is not substantially influenced by cellular RBPs." However, this conclusion appears to be at odds with their justification for using non-physiological K^+ concentrations in vitro. In their response to Reviewer 1, the authors argue that high K^+ conditions in the absence of the full cellular context, particularly RNA-binding proteins (RBPs) such as DHX36 and hnRNPF, which are known to destabilize rG4s, may yield misleading insights into rG4 temperature sensitivity. If RBPs are indeed critical for reproducing physiologically accurate rG4 folding behavior under high K^+ conditions, then the assertion that RNA alone governs the thermoresponsive properties of the RBM3 rG4s at physiological K^+ warrants further scrutiny. Specifically, the limited temperature-dependent folding observed under high K^+ in vitro suggests that RBPs may play a more substantial role than acknowledged. The authors should reconsider whether the inclusion of their statement about RNA sufficiency is fully supported by the experimental evidence presented.
2. Results presented in responses to reviewers show that G4 ligand NMM have minimal effect on the PSI of CREs in various genes. This data should be presented in the manuscript and discussed as a caveat. This potential reasons for this, and its implications in should be clear to readers in the discussion.
3. The authors make a claim about their work establishing new therapeutic avenues for neurodegenerative disease, but their work focuses on acute brain injury.
4. The authors and editors should review certain details regarding the formatting of the manuscript very carefully. For example, Fig. 2h and Fig. 2i still do not seem to correspond to the information in the figure legend (i.e., hRBM3 and mRBM3 seem to be flipped), as was mentioned in the first set of revisions. Other such examples may not have been identified by reviewers and will require thorough and close evaluation. Most importantly, figures and should have adequate and corresponding figure legends and be cited in the proper places in the manuscript.

(Remarks on code availability)

Version 2:

Reviewer comments:

Reviewer #1

(Remarks to the Author)

(Remarks on code availability)

The authors have addressed my concerns and I am supportive of publication of the revise manuscript.

Point-to-point response

We thank the editor and reviewers for their insightful comments, which we address below. We
have conducted additional experiments and revised the manuscript according to these comments.
The amendments are indicated in green.

We would like to mention that, as we have been asked to increase clarity of the manuscript, we
will be happy to further work on this once the scientific content is fixed, based on input from
reviewers and editorial comments.

Reviewer #1 (Remarks to the Author):

This is an interesting manuscript where the authors investigate the role of intronic G4s as potential
thermometers to regulate alternative splicing in mammal cells in response to temperature changed.
G4s have been linked to splicing in the past and they are known to affect alternative splicing, the
novelty here is that the authors present evidence suggesting that the fluctuation between a folded
or unfolded state of a G4 regulated by changes in temperature might be leveraged to regulate
differential splicing as a response of temperature shock. The authors use mutational analysis, PDS
treatment and K⁺ alteration to prove this point. They provide some interesting insights presented
in this manuscript, but I am afraid that the data generated are all mainly indirect and not sufficient,
in the current form, to prove a direct cause/effect between G4-folding/unfolding with temperature
in cells to splicing alteration. The main limitation currently is that their biophysical data cannot
really be reconciled with the cellular evidence and there is a lack of evidence demonstrating
changes in the G4-landscape in living cells upon temperature changes. If the authors can address
these main issues I would be supportive of publication in Nature Communication. Below I
articulate specific points to be addressed:

- We thank the reviewer for his/her interest in our study and the generally positive feedback. We
have addressed each of the comments below.

1) The folding/unfolding demonstrated by CD in vitro is very modest and, moreover, performed
at K⁺ concentrations that are not physiologically relevant. I would strongly doubt that small
changes in the CD like the one presented are representative of a significant fluctuation of
folded/unfolded G4 at 50mM KCl in such a small and low temperature range, given that especially
RNA G4s under physiological K⁺-concentrations are highly stable with melting temperatures above
70 C. The authors use NMR to further prove this point, but it looks like no K at all was used for
NMR experiments from the experimental section and therefore this cannot be considered
representative of how the actual fluctuation of the structure under physiological conditions with
more than 100mM K concentration would look like. An ideal experiment to address this would be
to perform in vitro splicing on a construct containing or not the G4 at different temperatures and
demonstrate that under physiological conditions the splicing is only affected when the G4 is
present but not in a control template.

-We appreciate the reviewer's suggestion regarding these experiments.

• The folding/unfolding demonstrated by CD in vitro is very modest; Small changes in the CD
like the one presented are representative of a significant fluctuation of folded/unfolded G4 at
50mM KCl in such a small and low temperature range.

For the presented CD spectrum data, the Y-axis represents the normalized ellipticity (10³
deg·cm²·dmol⁻¹), and the plotted curves reflect the mean values from independent triplicates rather

than a single representative dataset. For measurements at different temperatures, each sample was
equilibrated at the designated temperature for approximately 30 minutes before measurement (we
have added this description in the Methods) to ensure that the RNA folding/unfolding state was
stable at that temperature. Under high KCl (50 mM) conditions, the fluctuation of the
folded/unfolded G4 state is minimal, whereas under low KCl (0.1 mM) conditions, obvious
changes in the folded/unfolded state are observed. The curves at different temperatures are
consistent across triplicates, which very unlikely just reflect random fluctuations.

• *In vitro* CD experiment at not physiological K⁺ concentration and no K at all was used for
NMR experiment

Physiological K⁺ concentration is between around 120 to 150 mM, while in a cellular environment,
the folding/unfolding status of rG4s can be regulated by RNA-binding proteins (RBPs) (eg.
DHX36: Yuwei Zhang *et al*, Nat. Commun., 2024; hnRNPF: Cyril Dominguez *et al*, Nat. Struct.
Mol. Biol., 2010) beyond the intracellular rG4 stabilizer K⁺ concentrations. Thus, we assessed the
folding/unfolding dynamics of rG4s under both high (50 mM) and low (0.1 mM) K⁺ conditions in
CD spectroscopy. Interestingly, we observed that rG4 signals exhibited temperature sensitivity
only under low K⁺ conditions, not high K⁺ conditions, indicating that rG4s are less-temperature
sensitive if they are stabilized enough by their stabilizers, which is the possible reason that so far
no published studies have detect temperature sensitivity of rG4s under physiological temperature
fluctuations as they used too high concentrations of rG4 stabilizers in CD spectroscopy and NMR
*in vitro*. As we did not observe significant temperature-dependent folding/unfolding of rG4 under
high potassium conditions (50 mM), we did not test concentrations higher than 50 mM, such as
the physiological range (120 mM to 150 mM). We speculate that high KCl concentrations stabilize
rG4, making it less temperature-sensitive. Since NMR is more sensitive than CD, we used higher
KCl concentrations (2 mM) in NMR analysis to assess the temperature dependence of rG4. We
have added detailed information on KCl concentrations in the NMR experimental section for the
revision.

Regarding *in vitro* splicing assays, we have concerns about their feasibility in this context. *In vitro*
splicing is most efficient at 30°C, with reduced efficiency at 37°C, regardless of the presence of a
G4 structure (Krainer AR *et al*, Cell, 1984; Paul J. Furdon *et al*, 1986, PNAS; Mohammed
Albaqami *et al*, Plant Methods, 2018). Moreover, *in vitro* splicing may not adequately address
whether rG4s exhibit temperature sensitivity in a physiological or cellular environment.

Followed the reviewer's suggestions for the experiments, we performed minigene assays in live
cells. Specifically, we introduced the RBM3 G-rich element located at the 3' splice site (R1) and/or
5' splice site (R3) into an unrelated minigene (MINX) and a fluorescent reporter, and then
compared the wild-type (WT) sequence with its mutant or control backbone counterpart. This
approach allowed us to assess whether the WT G-rich element confers temperature sensitivity in a
cellular environment under physiological conditions. As expected, in the MINX minigene (**Fig.**
**R1a**), the WT sequence induced exon skipping at the low temperature (33°C) compared to that at
the high temperature (39°C). This pattern was significantly reduced in a reporter with the mutant
sequence (**Fig. R1b-c**). Moreover, we used another fluorescent minigene reporter (Huilin Huang
*et al.*, 2017, Genes Dev) to further confirm the temperature-sensitivity of R1 and R3 motifs of

RBM3 (Fig. R1d). The inserted cassette exon causes a frameshift in DsRed and does not contain
 a stop codon upstream of GFP, leading to GFP expression while preventing DsRed translation.
 However, if the cassette is skipped, DsRed is properly expressed, and the stop codon inhibits GFP
 production (Fig. R1d). Introduction of R1 and/or R3 around the splice sites of cassette exons into
 fluorescent minigene reporters significantly increased DsRed signal and decreased GFP signal at
 both 37°C and 39°C, leading to a higher DsRed/GFP ratio (Fig. R1e-f). This suggests enhanced
 cassette exon skipping and reduced inclusion after R1 and/or R3 introduction. These findings
 further confirm the repressive effect of G-rich elements on alternative splicing. Notably, the
 inhibitory effect was reduced at 39°C, indicating that the regulation by G-rich elements is
 temperature-sensitive.

 **Fig. R1 G-rich elements regulate temperature-sensitive alternative splicing.**

(a) Schematic of MINX minigene with R1 and R1 mutant motif of RBM3. A segment of the RBM3
 3a exon, including either the WT or mutated G-rich element were cloned at the 3' splice site of the
 MINX gene.
 (b) Representative splicing results of MINX minigene (a) by radioactive RT-PCR. The minigenes
 are transfected into HEK293T cells overnight, followed by different temperature treatments for
 another 24hrs.
 (c) Quantification of (b).
 (d) Schematic of the fluorescent minigene reporter containing R1 and/or R3 G-rich elements. The
 R1 and/or R3 elements were cloned near the splice sites of the cassette exon. The inserted cassette
 exon introduces a frameshift in DsRed and lacks a stop codon upstream of GFP, resulting in GFP
 expression but no DsRed expression. However, if the cassette is skipped, DsRed is expressed,
 and the stop codon prevents GFP expression (Huilin Huang *et al.*, 2017, Genes Dev). R1R denotes the
 replacement of the sequence near the splice sites of the cassette exon in the minigene with R1R,
 while the others represent the insertion of R1 and/or R3 near the splice sites.
 (e) Representative confocal images of HEK293 cells transfected with different fluorescent minigene
 reporter (d). The minigenes are transfected into HEK293 cells overnight, followed by different
 temperature treatment for another 24hrs.

(f) Quantification of images in (e). The mean fluorescent intensity in each image was automatically
quantified by Fiji. The scatter dot plots are shown as the mean \pm SD. Statistical analysis was
performed using unpaired t test. ns denotes no significance, * denotes $P \leq 0.05$, ** denotes $P \leq$
0.01 , *** $P \leq 0.001$.

2) Similarly, in cells the authors provide compelling evidence of splicing being altered as a
function of PDS treatment, K-channel blockage at different temperatures, speculating that the G4
stabilisation provided by either PDS treatment or K justifies the different results obtained a
different temperature in the presence or in the absence of these perturbative agents. However, PDS
data are really context dependent, and the authors speculate that this is due to the different
interactome that PDS disrupt at different sites, which can be a factor. However, PDS is also a DNA
damaging agent and is hard to disentangle the effects driven by G4 targeting to response to DNA
damage. Moreover, the authors have no evidence like immuno fluorescence for example to
demonstrate changes in G4 prevalence at different temperatures and under different conditions.
Hence, it hard to reconcile with confidence that changes in temperature affect splicing in a G4-
dependent fashion.

To address this it would be essential to have splicing data in cells with more than one G4 ligand
(i.e. commercially available PhenDC3 and NMM), to rule out any possible ligand-limited effect
(i.e. DNA damage for PDS). Evidence of differential G4 formation at identified sites by
immunoprecipitation or global changes by immuno fluorescence would also be ideal to
demonstrate a G4-dependent splicing effect.

-We appreciate the reviewer's concerns, which has helped enhance the strength of our evidence.

Following the reviewer's suggestions, we used two more G4 ligands (PhenDC3 and NMM) to rule
out ligand-limited effects in HEK293 cells. Several temperature-sensitive alternative cassette
exons (IQSEC, CDK4, and RBM3) were tested. Phen-DC3 treatment significantly reduced the
inclusion of these exons (**Fig. R2a-f**) across the tested temperatures, whereas NMM had no effect,
which is possibly due to differential responses of various stabilizers to different rG4 structures.
Further analysis using the RBM3 minigene (**Fig. R2g-h**) and its mutant (**Fig. R2i-j**) confirms that
the inhibitory effect of Phen-DC3 is also mediated by the G-rich elements R1 and R3.

**Fig. R2 G4 ligands regulate alternative splicing of temperature sensitive rG4 containing**
 **exons in HEK293 cells.**

**(a), (c), (e), (g), (i)** CRE inclusion level of CDK4 (a), IQSEC (c), RBM3 (e), RBM3 minigene (g)
 and RBM3 mutants (j) treated with DMSO or others (10 μ M) at different temperatures for 48 hrs
 (n=3-5) in HEK293 cells. A representative gel image is shown. PCR products and sizes are
 indicated on the right.

**(b), (d), (f), (h)** and **(j)**. Quantification results of a, c, e and g respectively. PSI indicates percentage
 of spliced-in. The scatter dot plots are shown as the mean \pm SD. Statistical analysis was performed
 using unpaired t test. ns denotes no significance, * denotes $P \leq 0.05$, ** denotes $P \leq 0.01$, *** P
 ≤ 0.001 .

In addition to G4-ligand experiments, we also performed G4-specific immunostaining and in-cell
 SHAPE-Map probing (See response to Reviewer 2, **Fig. R5** and **R6**) to assess differential rG4
 folding under various conditions. Exposure to low temperature (33°C) significantly enhanced the
 rG4-specific signal, as shown in **Fig. R3**, supporting the notion that rG4 folding is enhanced at
 lower temperatures in live cells.

**Fig. R3 Cold temperature substantially enhances global rG4 signal in HEK293 cells, as**
 **detected by G4-specific immunostaining.**

(a), (c) and (e) Representative immunostaining images of HEK293 cells at different temperatures
 and treatments in HEK293 cells. Cells were treated at different temperatures for 48hrs, followed
 by immunostaining with G4 specific antibody and confocal imaging.

(b), (d) and (f) Quantification of a, c and e with Fiji software. Quantifications were based on three
 replicate images, with signal intensity measured in 18-32 cells per image. Each dot represents the
 average intensity of a single cell. The scatter dot plots are shown as the mean ± SD. Statistical
 analysis was performed using unpaired t test. ****P ≤ 0.0001.

Moreover, we use voltage-gated potassium channel blocker (4-AP) to increase intracellular
 potassium in the glutamate-depolarized/-excited HT22 cells, and found 4-AP also significantly
 increased the global G4 signal in these cells (Fig. R4), indicating that increasing intracellular
 potassium promotes rG4 formation in cellular environments.

**Fig. R4 The potassium channel blocker AFP (Amifampridine) enhances global G4 signaling**
 **in glutamate-stimulated HT22 mouse neuroblastoma cells**

**(a)** Representative immunostaining images of HT22 cells at different treatments. Cells were treated
 with glutamate or glutamate+AFP for 24hrs, followed by immunostaining with G4 specific
 antibody and confocal imaging. Glutamate treatment can polarize neurons, activating the voltage-
 gated potassium channel and also can mimic the brain injury (Ke Lai *et al*, Nature 2024).

**(b)** Quantification of a with Fiji software. The scatter dot plots are shown as the mean ± SD.
 Statistical analysis was performed using unpaired t test. ****P ≤ 0.0001.

Providing that the authors can address experimentally these issues I would be supportive of
 publication in Nat Commun.

Reviewer #2 (Remarks to the Author):

Zhang et al. discovered a significant enrichment of rG4 across splice sites of cold-repressed exons.
In both human cells and plants, rG4s were previously found to be enhanced in response to cold
stress (Kharel et al., 2023 ; Yang et al., 2023). Thus, it is critical in determining the folding status
of rG4 *in vivo*. The authors need to determine the rG4 folding in different temperatures and
with/without potassium channel blocker using either DMS or SHAPE-based chemical probing
methods (Kharel et al.,2023).

- We acknowledge the reviewer's comments regarding the enhanced rG4 folding in response to
cold stress. However, how these folding is integrated into downstream gene regulation program
remains unclear. Particularly, as mammalian cells stay at a small range of temperature fluctuations,
the temperature sensing at molecular level for mammalian cells and plants might be quite distinct.

In this study, we demonstrate that RNA in mammalian cells can directly sense subtle temperature
changes through rG4 folding around splice sites, which convert temperature fluctuation signals
into gene expression program. Hence, for the first time, our study proposed that rG4s could serve
as thermosensors in mammalian cells.

We appreciate the reviewer's concerns regarding rG4 folding *in vivo*, which were also raised by
Reviewer 1 and 3. By integrating your suggestions and the Reviewer 1's suggestions, we have
conducted G4-specific immunostaining (suggested by Reviewer 1) and SHAPE-based probing
(suggested by Reviewer 2) experiments. These results demonstrated the temperature-sensitive rG4
folding in cellular environment. While G4 immunostaining results at different temperatures and
the potassium channel blocker treatment are presented in **Fig. R3 and Fig. R4** (see response to
Reviewer 1's point #2), SHAPE-based results were shown below (**Fig. R5**).

Additionally, we performed SHAPE-MaP probing to measure rG4 folding at different temperatures
in cellular environment (**Fig. R5a**). In this method, NAI (2-methylnicotinic acid imidazolide)
preferentially modifies the 2'-hydroxyl group of the ribose sugar in flexible regions of RNA, which
lead to the loop nucleotides connecting the G-tetrads, bulged residues within the G4 structure and
unstructured regions adjacent to the G4 core are modified, but not stable base-pairing or G-tetrad
stacking region (Junjie U. Guo and David P. Bartel, Science, 2016; Kwok, C. K. *et al*, Angew
Chem Int Ed Engl, 2016). The NAI-induced modification will form acylated adducts at modified
sites leading to mutations or indels in the cDNA during reverse transcription with Mn²⁺ using
Superscript II reverse transcriptase (Nathan A Siegfried *et al*, Nat. Methods, 2014; Matthew J
Smola & Kevin M Weeks, Nat. Protoc, 2018), and TIDE software was used to quantify the indel
percentage of the sanger sequencing results (**Fig. R5b**). Our results demonstrated that treatment
with the G4 ligands PDS and Phen-DC3 significantly increased the indel frequency surrounding
G-rich motifs near splice sites in several cold-repressed exons (CDK4, RBM3, and FKBP15)
containing putative rG3 near splice sites, suggesting that these motifs can form rG4 structures in
living cells. More importantly, the indel percentage was significantly enhanced at low temperatures
(33°C) (**Fig. R5c-f and Fig. R6**), supporting the notion that these G-rich elements form more stable
rG4s under cold conditions. Additionally, treatment with the potassium channel blocker AFP
(Amifampridine) or 4-AP further increased the indel percentage in the tested G-rich elements of
RBM3 cold-repressed exon3a, confirming their stabilizing effect on rG4s in living cells (**Fig. R5c-**
**f**).

Taken together, these results provide strong evidence that G-rich motifs around splice site indeed
 form more stable rG4s structure in living cells at low temperature or with G4 stabilizer treatment,
 suggesting it can function as physiological thermosensors.

 **Fig. R5 SHAPE-Map probing detects the temperature sensitive rG4 structure of the cold-**
 **repressed exons**

(a) Schematic of SHAPE-MaP probing method, HEK293 cells were treated with G4 stabilizers
 (PDS or Phen-DC3) or DMSO for 24 hours or incubated at different temperatures for 48 hours.
 Cells were then treated with 50 mM NAI, followed by DTT incubation. To get the enriched pre-

mRNA, chromatin-associated RNA was extracted, reverse-transcribed in an Mn²⁺-containing
 buffer, followed by cDNA amplification, gel-purification, and analysis via Sanger sequencing.
 **(b)** Schematic of the region of interest in SHAPE-Map using RBM3 R1 and R3 as examples.
 Chromatin-associated RNA was extracted and reverse-transcribed into cDNA in an Mn²⁺-
 containing buffer using Superscript II with a gene-specific primer binding ~50 bp downstream of
 the putative rG4. The cDNA was amplified with a high-fidelity polymerase and primers targeting
 ~50 bp upstream and downstream of the putative rG4. Indel percentages were analyzed using TIDE
 within the ±50 bp region of the putative rG4 (Brinkman *et al*, Nucl. Acids Res, 2014).
 **(c)** and **(d)** Representative sanger sequencing results of gel-purified SHAPE-modified products
 near the 5' (RBM3-R1) and 3' (RBM3-R3) splice sites of RBM3 exon 3a.
 **(e)** and **(f)** Quantified indel percentage of sanger sequencing data from b, d and f. (n=3-6). The
 indel percentages were quantified using TIDE software within the 50 bp upstream and downstream
 regions of the putative rG4. The scatter dot plots are shown as the mean ± SD. Statistical analysis
 was performed using unpaired t test. * denotes P ≤ 0.05, ** denotes P ≤ 0.01, ***P ≤ 0.001 and
 **** denotes P ≤ 0.0001.

 **Fig. R6 SHAPE-Map probing detects the temperature sensitive rG4 structure of the cold-**
 **repressed exons**

**(a)** and **(c)** Representative sanger sequencing results of gel-purified SHAPE-modified products
 near the splice sites of cold-repressed exon containing putative rG4 in CDK4 and FKBP15.
 **(b)** and **(d)** Quantified indel percentage of sanger sequencing data from a and c. The indel
 percentages were measured by TIDE software (Brinkman *et al*, Nucl. Acids Res, 2014) within the
 50 bp upstream and downstream regions of the putative rG4. The scatter dot plots are shown as
 the mean ± SD. Statistical analysis was performed using unpaired t test. * denotes P ≤ 0.05, **
 denotes P ≤ 0.01 and ***P ≤ 0.001.

Reviewer #2 (Remarks on code availability):

NA

Reviewer #3 (Remarks to the Author):

In this study, Zhang and colleagues investigate the significance of RNA G-quadruplex (rG4s)
enrichment around alternatively spliced temperature-sensitive cassette exons. They posit that the
cold-induced stabilization of rG4s surrounding splice sites prevents the inclusion of cassette exons
into the mature mRNA transcript. Using the neuroprotective RBM3 gene as a model, they suggest
that the temperature-dependent stabilization of rG4s around a poison exon 3a prevents the
inclusion of said exon in the mature transcript, preventing it from undergoing non-sense mediated
decay (NMD). They also determine that treatment of cells with potassium, an endogenous rG4
stabilizer, can mimic the effects of low temperature on alternative splicing and thus increase RBM3
expression at normothermia. Using a cellular and in vivo model of subarachnoid hemorrhage, they
test the hypothesis that 4-AP, an FDA-approved potassium channel blocker, can mimic the effects
of therapeutic hypothermia by increasing intracellular potassium concentrations and thus
expression of RBM3, affecting cellular outcomes in the brain and animal behavior.

Although the experiments in this study are generally well-conceived, this study needs significant
improvement to definitively reach the conclusions proposed and to convey the results adequately:

Major Points:

• The connection between therapeutic hypothermia, RBM3, subarachnoid hemorrhage, potassium,
and rG4s is not well-explained in the introduction section. A few of these concepts are explained
thoroughly (e.g., therapeutic hypothermia & subarachnoid hemorrhage) but seem disparate in their
presentation and the reader is left to infer the connection. The focus of the introduction should be
refined substantially. Certain details are excessive, such as the current state of treatment for
subarachnoid hemorrhage, and do not contribute significantly to understanding the study.

- We thank the reviewer for pointing out this problem. We have modified the introduction, made
it more focused and removed the current state of treatment for SH according to the reviewer's
suggestions.

- We have also added the information about RBM3 and its connection with theprapeutic
hypothermia, subarachnoid hemorrhage, connections between rG4 and potassium.

• The premises of experiments and descriptions of results need to be explained or clarified prior to
results being presented. For example, the relevance and role of potassium is unclear until much
later in the results. Similarly, the concept of glutamate stimulation mimicking brain injury is
suddenly introduced into the results text with no supporting context, background, or citations.
There are several other such instances throughout the manuscript that need to be carefully reviewed
by the authors, restructured, and presented in a logical manner.

- We apologize for the confusion caused by the content lacking a clear logical structure. We have
carefully reviewed the manuscript and modified these contents based on the reviewer's comments.

• Figures need to be reorganized significantly. Certain panels do not convey useful information for
understanding the results (e.g. Figs. 1a, 1b, S1a, S1b, S1f, and others). Figure legends in many

cases are not adequate and should describe all features of the corresponding figure (i.e.,
abbreviations, axes, annotations of gels and datapoints on panels, etc.); please carefully review
figure legends to ensure the descriptions are accurate and correspond to the proper panel, for
example in Fig. 2 panels H and I seem to be flipped. In addition, Fig. 7 which presents the working
model of this study would be better presented as an illustration or graphic in the discussion section,
not the results section.

- We thank the reviewer for these comments, which helped us improve the manuscript. Based on
the comments, we have removed Figs. 1a, S1a, S1b and so on to make the results easier to
understand. Regarding figure legends, we have modified the legends in the revised manuscript. As
suggested by the reviewer, we also move the Fig. 7 (now Fig. 8) into the discssuon section.

• In Fig. 1f and Fig. 1g, the authors show there is a significantly higher G-content and greater
presence of G4 motifs at specific regions surrounding cold-repressed exons (CREs). However,
there is an apparent difference in G-content and G4 motifs between cold-induced exons (CIEs)
and non-temperature sensitive exons (NTs), but these differences have not been statistically tested
and/or presented. Furthermore, Fig. 1h, Fig. 1i, and Fig. 1j also support the depletion of G-content
and G4 motifs surrounding CIEs. The manuscript focuses on CREs but does not adequately explain
the relevance of this depletion surrounding CIEs or mention its relevance in the proposed
mechanism.

-We appreciate the reviewer's interest in CIEs, which show depletion of G-content. However,
compared to CREs, the consistency of CIEs is not as good as that of CRE across different cell lines
as shown in Fig. 2b. Moreover, it's much more straightforward to study enrichment than depletion.
Hence, we mainly focus on the enriched G4 in CREs in this study. Regarding the relevance of G
motif depletion surrounding CIEs and its relevance in our proposed mechanism, such exons may
be regulated through different mechanisms, such as temperature-sensitive CLK activity (Haltenhof
*et al*, Mol Cell, 2020), although it is not clear, how that would be related to altere d G content.

• The premise of 4-AP treatment seems to be that it can be used to mimic the effects of therapeutic
hypothermia on neuroprotective RBM3 expression. The study should establish that RBM3 is fact
induced in response to therapeutic hypothermia in the brain and has similar effects on
cellular/behavioral outcomes following subarachnoid hemorrhage. Based on the current results, it
is unclear whether the effects of 4-AP at the tested dose induces changes in RBM3 expression that
are similar or more/less extreme than would be induced by therapeutic hypothermia.

- We thank the reviewer for raising the question reagrding RBM3 expression changes during
therapeutic hypothermia and in 4-AP treatment.

According to published studies, therapeutic hypothermia increases RBM3 protein levels by
approximately 1.8-fold *in vivo* in WT and young prion disease mouse hippocampus, and 4-fold *in*
*vivo* in the 5XFAD disease mouse hippocampus (Diego Peretti *et al*, 2015, Nature). Additionally,
RBM3 protein levels increase by approximately 3-fold *in vivo* in WT mouse brain and in cultured
neurons after low temperature treatment (for 3 days for mouse brain and 2 days for cultured
neurons) (Paulo Ávila-Gómez *et al*, Bain Comm., 2020; Sophorn Chip *et al*, Neurobiol Dis, 2011).
Furthermore, RBM3 mRNA expression increases by approximately 2-fold in cultured primary
neurons following 3 days of low temperature treatment (T C Jackson *et al*, Neuroscience, 2015).

Taken together, therapeutic hypothermia usually induces 1.8 to 4-fold increase of RBM3
expression.

On the other hand, our data show that 24-hour 4-AP treatment leads to approximately 1.5-fold
increases in RBM3 mRNA and 3.5-fold increases in protein levels in cultured SAH mouse
hippocampal HT22 cells. In the *in vivo* SAH mouse model, 4-AP treatment increases RBM3
mRNA levels by approximately 1.3-fold and protein levels by 1.7-fold, and knock-down of RBM3
abolished the neuroprotective effect of 4-AP. Therefore, the effects of 4-AP at our tested dose
induced quite similar RBM3 expression changes compared to therapeutic hypothermia.

We have revised the manuscript accordingly by adding this comparison result in the discussion.

• The sample size for several assays needs to be increased. The current sample size of n=3-4 used
in many of the *in vivo* readouts is not sufficient, especially given well-known animal to animal
variations combined with a treatment paradigm that necessitates a much larger sample size for
appropriate analysis. This is particularly important when quantifications are made based on
histochemical assays and/or when standard deviations are high (such as in Fig. 4K, Fig. 5b, Fig.
5i, etc).

For the formal *in vivo* mouse experiments, excluding preliminary studies, we used 10 mice per
treatment group, as shown in the behavioral test data. This sample size is sufficient for *in vivo*
mouse assays (Ke Lai *et al*, 2025, Nature).

As we specifically focus on the bleeding area, only a limited amount of brain tissue is available
from each mouse, which is marked as the area of interest in **Fig. 7b**.

To address the reviewer's concerns regarding the sample size in the *in vivo* readout experiments,
we have added two additional mouse brain samples, which increased the sample size of staining
experiments to 5 mice at least (sample size for NeuN staining is 10 as we also perform NeuN
staining in the samples for TUNEL staining.). The pattern we observed in the previous analysis is
confirmed, also with after increasing the sample size (**Fig. R7** and **Fig. R8**). The difference across
different conditions is obvious and significant with taking animal-to-animal variations into account.

**Fig. R7 4-AP mitigates neuronal damage in a mouse model of subarachnoid hemorrhage**
 **(SAH).**

**(a)** RBM3 protein expression of the SAH injected with lenti-shRBM3 and lenti-NC and sham
 mouse model in vivo after 4-AP and control administration, shown by RBM3 signal per cell in
 RBM3 immunostaining (n=5 mice, RBM3 signal of 10 neurons was quantified for each mouse
 slide).

**(b)** Percentage of apoptotic cells of the mice treated as **(a)** (n=5 mice).

**(c)** Neuronal count of the mice treated as a. Data was quantified from NeuN signal-positive cells
 (n=10 mice).

**(d)** Spongiosis score of the mice treated as a, quantified from hematoxylin and eosin (HE) staining
 (n=5 mice).

**(e)** The healthy neuronal counts in HE staining of the mice treated as **(a)** (n=5 mice).

**(f)** Quantified Nissl substance-positive cell counts with normal morphology (n=5 mice). **Fig. R7a**

and **e** are the scatter dot plots are shown as the mean ± SD. **Fig. R7c, d** and **f** are presented as box

and whisker plots with min to max. **Fig. 7b** is presented as a violin plot showing all points of

biological replicates. Statistical analysis was performed using two-way ANOVA with Tukey post

hoc correction. ns denotes no significance, * denotes $P \leq 0.05$, ** denotes $P \leq 0.01$, *** $P \leq 0.001$

and **** denotes $P \leq 0.0001$.

**Fig. R8 OE RBM3 mitigates neuronal damage in a mouse model of subarachnoid**
 **hemorrhage (SAH).**

(a) Spongiosis score of the mice treated with OE-NC or OR-RBM3 in SAH and Sham, quantified
 from hematoxylin and eosin (HE) staining (n=5 mice).

(b) The No. of healthy neurons in HE staining of the mice treated as (a) (n=5 mice).

(c) Quantified No. of Nissl substance-positive cells with normal morphology of the mice treated
 as (a) (n=5 mice). **Fig. R8a** and **c** are presented as box and whisker plots with min to max. **Fig.**
 **R8b** is the scatter dot plots are shown as the mean \pm SD. Statistical analysis was performed using
 two-way ANOVA with Tukey post hoc correction. ns denotes no significance, * denotes $P \leq 0.05$,
 ** denotes $P \leq 0.01$, *** $P \leq 0.001$ and **** denotes $P \leq 0.0001$.

• The formation of rG4 structures in the mouse brain is not completely convincing. The
 establishment of rG4 folding seems to be based largely on in-silico analysis. While the authors do
 perform CD/NMR, the folding of the tested fragments outside of the cellular environment is not
 direct evidence of rG4 folding in the cellular environment, let alone within the brain. Specifically,
 the use of short fragments in CD/NMR precludes the consideration of the extended sequence of
 the full-length mRNA, or cellular features, that could affect folding and rG4 formation in vivo.
 Also, the authors attempt to further prove rG4 formation using rG4-seq data from another study
 that was conducted in HeLa cells. The reader is led to believe that this extends to the formation of
 rG4s in the brain, however they do not present a direct validation of this in the current study. The
 entire premise of the mechanism is based on rG4 formation, so biochemical assays to confirm this
 folding in the brain are critical. Based on the current data, rG4 formation in the brain cannot be
 assumed. Further experiments, including rG4-seq or SHALiPE may be used to characterize these
 structures and their presence in the native environment, which would lend credence to their
 existence and relevance in the brain and in this particular model.

- We thank the reviewer for concerns about rG4 folding in the cellular environment and in the
 brain. We acknowledge the limitation of the previous data in directly proving the rG4 folding
 within the cells. This is also a concern of the Reviewer 1 and 2. We have now included more data
 showing rG4 folding in cellular environments (see our response to Reviewer 1 and 2).

• Given the therapeutic relevance of 4-AP, dose-dependent effects on RBM3 expression and
 phenotypic/cellular outcomes are critical. A justification for the current dose is not described,
 neither is a dose-dependent characterization of the aforementioned readouts presented in this study.

- The dosage used in our *in vivo* experiments was based on two previously published studies:
 Franciosi *et al.*, 2006 (J. Neurosci) and Dietrich *et al.*, 2021 (Neurol Neuroimmunol

Neuroinflamm). We have explicitly referenced these studies in the Methods section of our
manuscript. To make it more clear to readers, we have now revise the manuscript by adding the
following description:

“The dosage of 4-AP in mice was based on previously established protocols to ensure effectiveness
while minimizing significant side effects”

Minor Points:

• The results section begins with a summary of a previous study. It is unclear which findings are
specific to the current study. If the results in the first paragraph are taken directly from a previous
study, they should be mentioned only saliently and as needed in the results section or summarized
in the introduction. Previous results should not be associated with figures unless previous data is
being reanalyzed in a novel way.

- We have re-structured the results in the revised manuscript to avoid potential confusions.
• Figures and panels should be referred to in the order in which they appear throughout the
manuscript.

- This has been checked, thank you.

• The lack of clarity about definitions/abbreviations, naming conventions, and pertinent
methodological details in the results section severely hinders the readability of the results. For
example, four regions are delineated with undefined abbreviations in Fig. 1C but not explained in
the results or the legend. There are several other examples of this, including the naming convention
for the mutants/minigenes.

- We have add definations/abbreviations.

• Given that several cell types were used in this study, they should be consistently mentioned for
clarity (e.g., in the hemin-induced model results, it is unclear what cell type is being used until
well into the corresponding text).

- We have added the corresponding information.

• The authors make several claims in the results that are based on overinterpretation or speculation.
For example, they establish the conservation of G-enrichment in cassette exons broadly in
mammals and suggest that this implies a role in splicing events during hibernation (presumably to
establish a connection with temperature regulation?), although it is widespread in mammals that
do not hibernate, thus one is confused about the link between G-enrichment and hibernation as it
could serve a function that is independent of hibernation. Separately, they suggest that increased
intensity of Nissl staining implies increased protein synthesis, which is not substantiated by other
confirmatory assays. These points may be expanded upon in the discussion but should be removed
from the results.

- We have deleted the hibernation section and the correlation between Nissl staining and increased
protein synthesis.

• Statements that make claims not supported by data should be accompanied by citations (multiple
in some cases). Several claims throughout the manuscript are not qualified with citations.

- We appreciate the reviewer for raising this point, and we have included the relevant citations in
the revised manuscript.

• The mechanism proposed is posited as an ‘alternative mechanism’ to CLK-dependent alternative
splicing, however, their results show that G-enrichment around exons occurs in both CLK-
dependent and CLK-independent spliced genes. This is confusing and I am not convinced that this
is necessarily an alternative mechanism.

- We thank the reviewer for raising this concern and appologize for the confusion. To increasing
the clarity, we have now removed the results related to CLK in the revised manuscript.

**Point-to-point response**

**Reviewer #1 (Remarks to the Author):**

The authors have now addressed most of the technical concerns I have raised and I am now
supportive of publication in Nat Commun.

The only minor detail to fix prior to publication is to specify the antibody used for G4 IF (BG4),
this is mentioned in the methods but not in the figure caption (please add this for clarity). There
are many G4-specific antibodies and the authors need to be more explicit on what they have used.
Also the seminal papers where the antibody has been used and developed should be cited.

We sincerely appreciate the reviewer's constructive feedback and overall positive comments
throughout the review process. In response to this point, we have added the full information for
the BG4 antibody in the relevant figure legends. Additionally, we have cited key references where
this antibody was previously used (Deiana *et al.*, *Nucleic Acids Res.* 2023 and Wulfridge *et al.*,
*Mol Cell*, 2023) as well as the original study that developed the antibody (Biffi *et al.*, *Nat Chem*,
2013). These citations have been included in the Results, Methods, and Figure legends wherever
the antibody is mentioned, to ensure proper context and attribution.

**Reviewer #2 (Remarks to the Author):**

The authors did not correctly perform the experiments to address my comment. The SHAPE-based
method captures the preferential modification of NAI on the last G of every G tract in response to
potassium or rG4 ligand. Either SHAPE-induced reverse transcription stalling or mutation occurs
at the last G of every G tract. This modification pattern is very specific. The indel surrounding the
G-rich motif is not specific to rG4. It is more likely due to the single-stranded nucleotide modified
by NAI. Alternatively, the authors could also use the DMS-based method described in Kharel *et al.*,
2023, which is based on the DMS modification on the N7G position, where rG4 formation
prevents DMS modification of the N7G position.

We thank the reviewer for this thoughtful and detailed feedback, which has helped us to further
refine and clarify our experimental rationale and interpretation. Below, we respond point-by-point:

1. "The authors did not correctly perform the experiments to address my comment."

In response to the reviewer's initial suggestion, we performed SHAPE-MaP (Selective 2'-
Hydroxyl Acylation analyzed by Primer Extension and Mutational Profiling), a well-established
*in vivo* SHAPE-based chemical probing method (Siegfried, N. A. *et al.*, *Nat Methods*, 2014 and
Smola, M. J. *et al.*, *Nat Protoc*, 2018), to assess rG4 folding under different temperatures, the well-
known G4 stabilizer PDS, and Phen-DC3 as positive control. These experiments were designed to
test the temperature sensitivity of rG4 folding in a cellular context, as the reviewer recommended.

2. "The SHAPE-based method captures the preferential modification of NAI on the last G of every
G tract..."

We agree that SHAPE reagents, including NAI, target flexible regions of the RNA backbone.
However, SHAPE modification patterns in G-rich sequences depend strongly on the structural
context. In rG4 structures, the guanines participating in G-quartet formation are generally
protected from SHAPE reactivity, while loops and flanking single-stranded regions exhibit
increased reactivity signal (Kwok, C. K. *et al*, *Angew. Chem. Int. Ed. Engl.* 2016, Kwok, C. K. *et*
*al*, *Nature Methods*, 2016, Kwok, C. K. *et al*, *Cold Spring Harb Perspect Biol*, 2018 and Prakash
Kharel *et al*, *Nucleic Acids Res.* 2020). Moreover, the location of SHAPE reactivity can vary
depending on the topology of the G4 (e.g., parallel, antiparallel or hybrid) and the nature of
loop sequences (Kwok, C. K. *et al.*, *Nat Methods*, 2016), and is not necessarily confined to the
last guanine in each G-tract. While reverse transcription stalling or mutations may occur at the last
guanine of a G-tract, this is neither a universal nor exclusive signature of rG4 folding. Therefore,
the interpretation of SHAPE-MaP data for rG4 structures must be informed by a combination of
nucleotide-level reactivity, structural context, and chemical perturbation controls.

3. “The indel surrounding the G-rich motif is not specific to rG4...”

To strengthen our conclusions and control for nonspecific NAI effects, we performed SHAPE-
MaP under multiple conditions: varying temperature and treatment with rG4-stabilizing ligands
(PDS and Phen-DC3).

Our analysis included indel rates across ± 50 nt regions flanking predicted rG4 motifs. Importantly,
we observed a localized increase in SHAPE reactivity and indel frequency specifically following
rG4 ligand treatment, not in controls (See **Fig. 4c-f**, **Supplementary Fig. 7** and **Supplementary**
**Fig. 8**). These effects were consistent across two structurally distinct rG4 ligands and were
decreased when rG4 formation was presumably inhibited at higher temperature.

These data reflect a reproducible and structure-specific change, consistent with rG4 folding, rather
than random reactivity with single-stranded RNA. Furthermore, we consulted Prof. Chun Kit
Kwok (City University of Hong Kong), a recognized expert in RNA G-quadruplex biology and
chemical probing, who confirmed the appropriateness and interpretability of SHAPE-MaP for
detecting rG4 formation *in vivo*. His input is acknowledged in the revised manuscript.

4. “Alternative validation using DMS or other methods”

While we appreciate the suggestion to use the DMS-based approach targeting N7G as described
by Kharel *et al.* (2023), this method is technically challenging to implement in our current system
due to limitations in chemical access and biosafety regulations. Nevertheless, we acknowledge its
potential and have cited this work as an important complementary technique for future studies.

5) “Independent validation by rG4 immunostaining”

To further support our SHAPE-MaP results, we performed rG4 immunostaining using the well-
established BG4 antibody (e.g. used in Deiana *et al.*, *Nucleic Acids Research*; Wulfridge *et al.*,
*Mol Cell*, and Biffi *et al.*, *Nature Chem*) under the same experimental conditions. This method
provides an orthogonal, spatially resolved readout of rG4 formation *in situ*. The changes in BG4
signal across temperature and potassium conditions are consistent with our interpretation of the

SHAPE-MaP data, reinforcing the conclusion that rG4 folding is dynamically modulated in cells
by physiological stimuli.

In summary, we believe that the combination of temperature-dependent SHAPE-MaP profiling,
rG4 ligand perturbation, indel analysis, expert consultation, and orthogonal validation by BG4
staining together provide robust evidence supporting our conclusions about rG4 folding dynamics
*in vivo*. We respectfully suggest that our data directly address the reviewer's concern regarding
structural specificity and functional validation.

We thank the reviewer again for raising these important points, which have allowed us to
strengthen the clarity, rigor, and interpretation of our findings.

**Reviewer #2 (Remarks on code availability):**

N/A

**Reviewer #3 (Remarks to the Author):**

In their revised version of “Stabilizing a mammalian RNA thermometer confers neuroprotection
in subarachnoid hemorrhage” Zhang *et al.* provide compelling mechanistic evidence for the role
of RNA G-quadruplexes (rG4s) functioning as temperature-sensitive regulatory elements that
modulate alternative splicing. Their revision version incorporates several new assays, including
minigene reporter systems, rG4 immunostaining, and SHAPE-based techniques to assess the
stability of rG4 folding *in vivo*, addressing the primary scientific critique of all three reviewers in
their prior version. Together, these additions strengthen the claim that the structural stability of
rG4s near the splice sites of the poison exon 3a in RBM3 is modulated by changes in temperature
and intracellular potassium levels, thus governing RBM3's capacity for expression and eventual
translation. Given RBM3's established protective role in the brain, this study identifies a potential
therapeutic target for modulating outcomes in neurological injury.

The experimental evidence presented is strong, but we hope the authors consider refining the
discussion to acknowledge key caveats of their study/techniques and do a thorough evaluation of
certain details in their manuscript to ensure it is ready for publication.

Minor Points:

1. Of note, in the revised manuscript, the authors conclude that the RBM3 rG4 acts as an
autonomous thermosensor and that “the temperature range is not substantially influenced by
cellular RBPs.” However, this conclusion appears to be at odds with their justification for using
non-physiological K⁺ concentrations *in vitro*. In their response to Reviewer 1, the authors argue
that high K⁺ conditions in the absence of the full cellular context, particularly RNA-binding
proteins (RBPs) such as DHX36 and hnRNPF, which are known to destabilize rG4s, may yield
misleading insights into rG4 temperature sensitivity. If RBPs are indeed critical for reproducing
physiologically accurate rG4 folding behavior under high K⁺ conditions, then the assertion that
RNA alone governs the thermoresponsive properties of the RBM3 rG4s at physiological K⁺
warrants further scrutiny. Specifically, the limited temperature-dependent folding observed under

high K⁺ *in vitro* suggests that RBPs may play a more substantial role than acknowledged. The
authors should reconsider whether the inclusion of their statement about RNA sufficiency is fully
supported by the experimental evidence presented.

We appreciate the reviewer's thoughtful comments and the opportunity to clarify our interpretation.
We acknowledge that RNA-binding proteins (RBPs) can modulate rG4 folding dynamics in cells.
Indeed, this is precisely why we caution against interpreting rG4 folding behavior based solely on
*in vitro* experiments performed under high K⁺ conditions, which lack the cellular milieu—
including RBPs—that can destabilize rG4 structures.

However, our conclusion that RBM3 rG4 functions as an autonomous thermosensor is based
primarily on live-cell experiments. In these physiologically relevant contexts, we observed clear
and reproducible rG4 folding transitions in response to modest temperature changes, without any
direct manipulation of RBPs. This supports the idea that RNA itself possesses an intrinsic ability
to respond to temperature shifts, which we also see in our *in vitro* experiments containing only
RNA. Together, this leads to the concept of rG4s acting as thermosensitive elements in mammalian
cells.

We do not intend to exclude the potential influence of RBPs on the fine-tuning of rG4 folding *in*
*vivo*. Rather, our findings indicate that even in the presence of endogenous RBPs, rG4s can
undergo temperature-dependent structural changes. Thus, we believe that RNA alone is
sufficient—though not necessarily exclusive—in governing this thermoresponsive behavior under
physiological conditions.

We have revised the relevant text in the manuscript to clarify this interpretation and better reflect
the role of RBPs in modulating the RNA's intrinsic structural responsiveness to temperature.

2. Results presented in responses to reviewers show that G4 ligand NMM have minimal effect on
the PSI of CREs in various genes. This data should be presented in the manuscript and discussed
as a caveat. This potential reasons for this, and its implications in should be clear to readers in
the discussion.

We appreciate the reviewer's insightful comment. In response, we have now included the NMM
treatment data in the revised manuscript (see new **Supplementary Fig. 3a-h, Supplementary Fig.**
**4f-g** and corresponding results and discussion section). As shown, NMM has minimal impact on
the percent-spliced-in (PSI) values of cassette exons (CREs) harboring rG4 motifs, under the tested
conditions.

We agree that this observation warrants discussion, and we have addressed it in the revised
discussion section. Specifically, this may reflect the differential binding affinities of NMM for
rG4s, influenced by factors such as backbone orientation, loop architecture, and stabilizing
structural elements.

Taken together, our data support the model that RNA alone can function as a thermosensor in cells,
with RBPs playing a modulatory—rather than obligatory role in shaping temperature-dependent
splicing outcomes. We have revised the manuscript to clarify this interpretation and highlight both

the potential limitations of G4 ligands in functional assays and the importance of considering
cellular context when probing rG4 biology.

3. The authors make a claim about their work establishing new therapeutic avenues for
neurodegenerative disease, but their work focuses on acute brain injury.

We thank the reviewer for this important observation. We agree that the primary focus of our
current study is on acute brain injury and have clarified our language accordingly. In the revised
manuscript, we now state:

“Altogether, our results lay the groundwork for promising new therapeutic avenues for acute brain
injury that could benefit from elevated RBM3 expression, and more broadly, for conditions that
can be treated with hypothermia, such as neurodegenerative diseases (Peretti *et al.*, *Nature*, 2015).”

This revision more accurately reflects the scope of our data, while also acknowledging that our
findings may have broader implications. The rationale for referencing neurodegenerative diseases
stems from prior work showing that RBM3 is upregulated in response to cooling and plays a
neuroprotective role in models of chronic neurodegeneration (e.g., Peretti *et al.*, *Nature*, 2015).
Our study adds to this field by elucidating mechanisms by which RBM3 expression is temperature-
regulated at the RNA level. Therefore, while our experimental data are specific to acute injury
models, the molecular insights gained here may help to inform strategies applicable to a wider
range of hypothermia-related therapeutic contexts, including neurodegeneration.

We hope this clarification appropriately contextualizes our conclusions.

4. The authors and editors should review certain details regarding the formatting of the manuscript
very carefully. For example, Fig. 2h and Fig. 2i still do not seem to correspond to the information
in the figure legend (i.e., hRBM3 and mRBM3 seem to be flipped), as was mentioned in the first
set of revisions. Other such examples may not have been identified by reviewers and will require
thorough and close evaluation. Most importantly, figures and should have adequate and
corresponding figure legends and be cited in the proper places in the manuscript.

We thank the reviewer for this careful observation. We have corrected the Figure legend for **Fig.**
**2h** and **Fig. 2i** to accurately reflect the data, which now reads: “Exon 3a inclusion in the hRBM3
(**h**) and mRBM3 minigene (**i**) after G4 stabilizer PDS or control treatment in HEK293T cells.” We
sincerely apologize for the oversight.

In response to the reviewer’s broader concern, we have thoroughly re-examined the entire
manuscript, including all figure panels, legends, and corresponding citations in the main text. We
have made necessary corrections to ensure consistency, accuracy, and clarity throughout. We are
grateful for the reviewer’s diligence, which helped us improve the overall presentation and rigor
of the manuscript.